# Deficiency of Axl aggravates pulmonary arterial hypertension via BMPR2

Tatyana Novoyatleva [1✉], Nabham Rai[1], Baktybek Kojonazarov[1,2], Swathi Veeroju[1], Isabel Ben-Batalla[3,4], Paola Caruso [5], Mazen Shihan[1], Nadine Presser[1], Elsa Götz[1], Carina Lepper[1], Sebastian Herpel[1], Grégoire Manaud[6], Frédéric Perros [6], Henning Gall [1], Hossein Ardeschir Ghofrani [1], Norbert Weissmann[1], Friedrich Grimminger[1], John Wharton [7], Martin Wilkins [7], Paul D. Upton [5], Sonja Loges[3,4], Nicholas W. Morrell [5], Werner Seeger [1,8] & Ralph T. Schermuly [1✉]

Pulmonary arterial hypertension (PAH), is a fatal disease characterized by a pseudo-malignant phenotype. We investigated the expression and the role of the receptor tyrosine kinase Axl in experimental (i.e., monocrotaline and Su5416/hypoxia treated rats) and clinical PAH. In vitro Axl inhibition by R428 and Axl knock-down inhibited growth factor-driven proliferation and migration of non-PAH and PAH PASMCs. Conversely, Axl overexpression conferred a growth advantage. Axl declined in PAECs of PAH patients. Axl blockage inhibited BMP9 signaling and increased PAEC apoptosis, while BMP9 induced Axl phosphorylation. Gas6 induced SMAD1/5/8 phosphorylation and ID1/ID2 increase were blunted by BMP signaling obstruction. Axl association with BMPR2 was facilitated by Gas6/BMP9 stimulation and diminished by R428. In vivo R428 aggravated right ventricular hypertrophy and dysfunction, abrogated BMPR2 signaling, elevated pulmonary endothelial cell apoptosis and loss. Together, Axl is a key regulator of endothelial BMPR2 signaling and potential determinant of PAH.

[1] Universities of Giessen and Marburg Lung Center (UGMLC), Excellence Cluster Cardio-Pulmonary System (ECCPS), Member of the German Center for Lung Research (DZL), Justus-Liebig-University Giessen, Giessen, Germany. [2] Institute for Lung Health, Giessen, Germany. [3] Department of Oncology, Hematology and Bone Marrow Transplantation with section Pneumology, Hubertus Wald University Comprehensive Cancer Center Hamburg, University Medical Center Hamburg-Eppendorf, Hamburg, Germany. [4] Department of Tumor Biology, Center of Experimental Medicine, University Medical Center Hamburg-Eppendorf, Hamburg, Germany. [5] Department of Medicine, University of Cambridge, Cambridge, UK. [6] Université Paris–Saclay, AP-HP, INSERM UMR_S 999, Service de Pneumologie et Soins Intensifs Respiratoires, Hôpital de Bicêtre, Le Kremlin Bicêtre, France. [7] Centre for Pharmacology and Therapeutics, Department of Medicine, Imperial College London, London, UK. [8] Max Planck Institute for Heart and Lung Research, Bad Nauheim, Germany.
✉email: tatyana.novoyatleva@innere.med.uni-giessen.de; ralph.schermuly@innere.med.uni-giessen.de

Pulmonary arterial hypertension (PAH) is a life-threatening disease characterized by progressive pulmonary vasculopathy usually culminating in right heart failure[1]. Vasoconstriction, endothelial dysfunction, structural remodeling of pulmonary arteries, and inflammation contribute to the progression of the disease[2], which shares several features with cancer[3].

Aberrant regulation and signaling of the TAM family of receptor tyrosine kinases (RTKs)—Tyro3, Axl, and Mer—are tightly linked with cancer progression, metastasis, and chemoresistance[4–8]. Axl is activated by binding to its sole ligand, growth arrest-specific protein 6 (Gas6)[9], and via ligand-dependent and -independent dimerization as well as heterodimerization with non-TAM receptors[10–12]. The physical association of AXL with particular RTKs, characterized by cross-linking co-immunoprecipitation, implicates AXL-mediated signaling synergy in the acquired resistance of several RTKs to specific receptor-targeted inhibitors[7,8,13]. Axl plays an essential role in vascular endothelial growth factor A (VEGF-A)/VEGF receptor (VEGFR) signaling[12]. In endothelial cells (ECs), Axl activation modulates anti-apoptotic effects, possibly through its association with specific integrin complexes[14]. Axl is highly expressed in vascular smooth muscle cells (SMCs), where it plays a protective role during vascular injury[15,16]. Upregulation and secretion of Gas6 with subsequent activation of Axl, is one of the hallmarks of vascular injury response and vascular remodeling. Gas6-induced Axl signaling contributes to the survival of human pulmonary arterial ECs (hPAECs)[15], increases SMC proliferation and migration, blocks apoptosis[17,18], and regulates SMC immune heterogeneity and extracellular matrix remodeling[19].

Both proteins have been reported to be upregulated and activated in various human malignancies, in which their expression is tightly correlated with poor prognosis and increased invasiveness/metastasis[20]. Inhibition of Axl kinase in multiple model systems reduces cancer cell survival, enhances chemosensitivity, and diminishes metastatic potential[6]. On another side, ablation of Axl, a key regulator of innate immune response[21], has been associated with increased production of proinflammatory cytokines and failure to clear neutrophils undergoing apoptosis, thus favouring a tumor-promoting environment[22,23].

We investigated the role of Gas6/Axl in PAH. Axl inhibition by the small molecule inhibitor R428 markedly reduced human pulmonary arterial SMC (hPASMC) proliferation and migration, but prompted toxic effects in hPAECs. These effects were confirmed in vivo, as R428 aggravated experimental pulmonary hypertension (PH) in rats. R428 elevated pro-inflammatory responses, exacerbated EC apoptosis and loss, and abrogated the bone morphogenetic protein receptor 2 (BMPR2) signaling pathway. Axl expression markedly declined in blood outgrowth endothelial cells (BOECs), in laser-assisted microdissected intrapulmonary vessels (MDVs), and in the pulmonary endothelium of IPAH patients. Axl deletion or blockage reduced BMPR2-related canonical SMAD signaling. Importantly, Axl and BMPR2 co-interacted in hPAECs, indicating a previously unrecognized association between these two receptors. Taken together, our studies demonstrate the protective role of Gas6/Axl in PAH and highlight a novel liaison between Gas6/Axl and BMP/BMPR2 signaling pathways.

## Results

### R428 inhibits proliferation and migration of PASMCs.
The expression of Axl was augmented in the lungs of Sugen 5416/hypoxia (SuHOX) and monocrotaline (MCT) rat models, and chronic-hypoxia mouse model of PH, as compared to controls (Supplementary Fig. 1). As Axl expression was strongly induced in pulmonary vessels (Supplementary Fig. 1g) and pulmonary arterial smooth muscle cells (PASMCs) in experimental PH (Supplementary Fig. 1h), we explored the functional effect of Axl inhibition on hPASMC proliferation and migration using a clinically tested small molecule inhibitor R428 (BGB324). Pretreatment with R428 dose-dependently inhibited growth factor-induced proliferation and migration of both control and IPAH hPASMCs (Fig. 1a–d and Supplementary Fig. 2a–d). An anti-proliferative effect of R428 was confirmed by western blot analyses using cell cycle markers Cyclin D1 and p27 (Fig. 1e, f), and by immunofluorescence staining using the proliferation marker Ki67 (Fig. 1g, h). Lentiviral overexpression of Axl in hPASMCs conferred a growth advantage and increased sensitivity to R428 compared with the expression of a control construct (Fig. 1i, j). Importantly, reduction of Axl expression by lentiviral short hairpin RNA abrogated sensitivity to R428 at 1 and 2.5 μM, indicating that the inhibitory effect is mediated via Axl and not by off-targets (Fig. 1i, k). Exogenous stimulation of Axl with recombinant Gas6 did not increase hPASMC proliferation (Supplementary Fig. 2e, f).

### R428 inhibits proliferation and induces apoptosis of PAECs.
Next we investigated the impact of R428 on human pulmonary arterial endothelial cells (hPAECs) in vitro. An apparent reduction in epidermal growth factor (EGF)-induced proliferation was observed for R428 pre-treated cells (Fig. 2a). Importantly, R428 induced both hPAEC cytotoxicity (Fig. 2b, c) and apoptotic cell death at the nanomolar range (Fig. 2d, e and Supplementary Fig. 3a). Remarkably, the IPAH-PAECs showed a stronger cytotoxic response in comparison to donor non-PAH PAECs (Fig. 2b, c). The cytotoxic effects in hPASMCs have been seen with high doses of R428 (Supplementary Fig. 3c, d). The R428 initiated apoptosis was strongly inhibited by Gas6 stimulation, supporting an anti-apoptotic role of Gas6 in vascular cells (Supplementary Fig. 3e).

Endothelial expression of adhesion molecules is one of the characteristic features of vascular dysfunction and damage[24,25]. Thus, next we aimed to investigate the effect of Axl blockage on markers of endothelial dysfunction and damage/vascular injury in hPAECs. R428 treatment and Axl knock-down each substantially increased mRNA levels of intercellular adhesion molecule 1 (ICAM1) (Fig. 2f, h). Interestingly, R428 treatment and Axl deletion provided opposing results on the expression of vascular cell adhesion molecule 1 (VCAM1), indicating that small molecule inhibitor R428 and Axl deletion might contribute to non-overlapping molecular pathways (Fig. 2g, h). R428 also initiated a dose-dependent increase in ICAM1 mRNA expression in hPASMCs, suggesting that Axl may regulate the immune activation of SMCs (Supplementary Fig. 3f). Furthermore, R428 treatment of cultured hPAECs resulted in a significant and time-dependent increase in endothelial monolayer permeability, as demonstrated through the use of fluorescein isothiocyanate-dextran (FITC-Dextran) transwell assays (Supplementary Fig. 3g).

### R428 aggravates experimental PH.
MCT and SuHOX rat models of PH developed evident RV hypertrophy (Fig. 3a, f). As compared to controls, animals receiving R428 had more severe RV hypertrophy (Fig. 3a, f), augmented right ventricular systolic pressure (RVSP) (Fig. 3b, g), impaired cardiac output (Fig. 3c, h), and worsened tricuspid annular plane systolic excursion (Fig. 3d, i), and a significantly increased total pulmonary vascular resistance index (Fig. 3e, j). No changes in the systemic arterial pressure and heart rate were noted after R428 application.

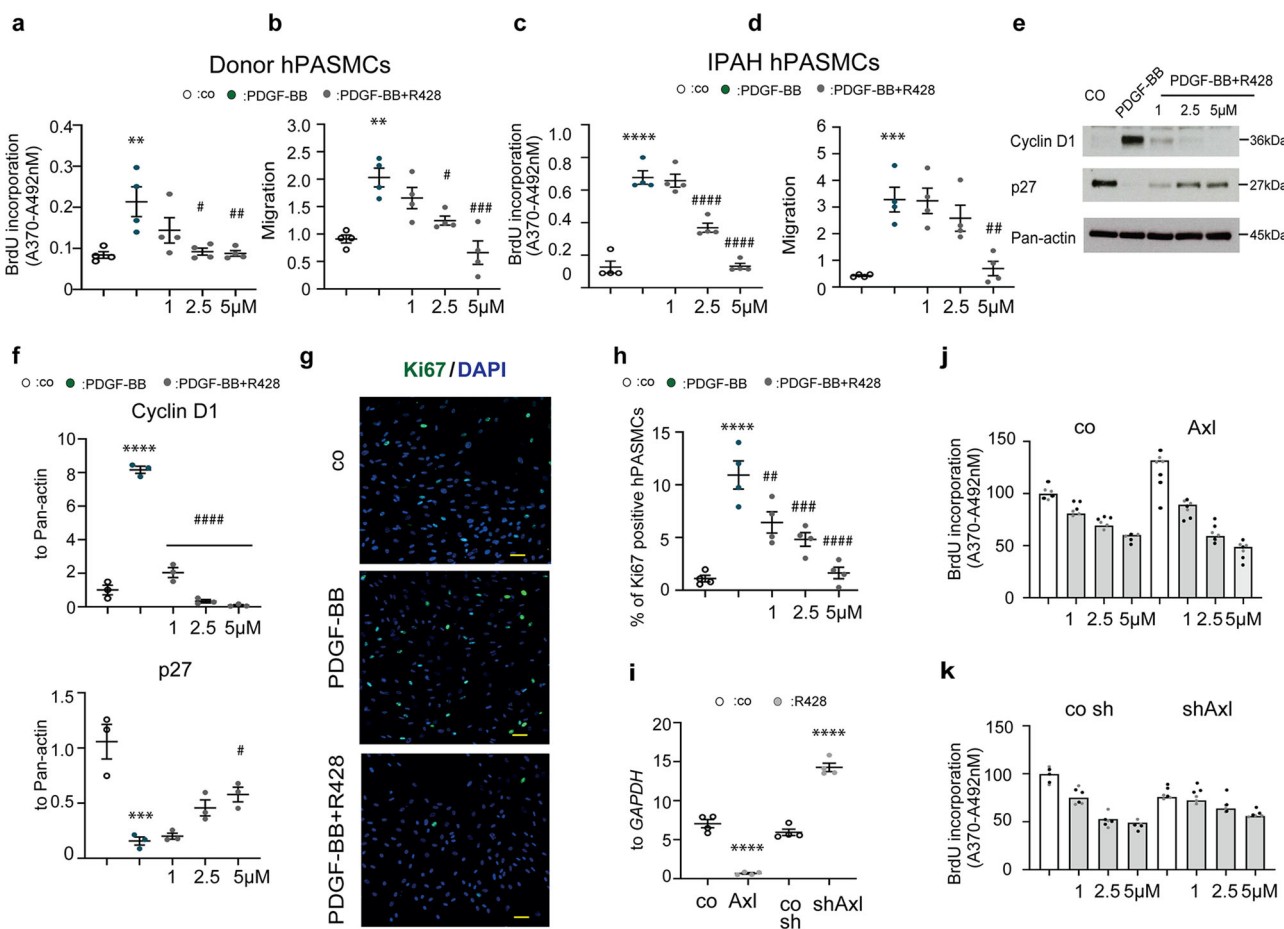

**Fig. 1 Axl regulates the proliferation and migration of human pulmonary arterial smooth muscle cell (hPASMC). a**, **c** HPASMCs from healthy individuals and patients with idiopathic pulmonary arterial hypertension (IPAH) exposed to 30 ng/ml platelet-derived growth factor-BB (PDGF-BB) with or without R428 at indicated concentrations. Control cells (co) were exposed to dimethyl sulfoxide (DMSO). Proliferation potential was assessed by 5-bromo-2′-deoxyuridine (BrdU) colorimetric enzyme-linked immunosorbent assay (ELISA). Assessment of DNA synthesis was performed four hours after BrdU addition [$A_{370nm}$–$A_{492nm}$]. Data from $n = 4$ biological independent experiments performed in triplicate are presented. **b**, **d** Migration efficiency was determined by transwell migration assays. Data from $n = 4$ biological independent experiments performed in triplicate are presented as the $n$-fold change normalized to DMSO-treated control cells. **e**, **f** Western blot images and densitometry quantification of Cyclin D1 and p27 expression in hPASMCs after R428 treatment, followed PDGF-BB stimulation. Pan-actin served as a loading control. Data from $n = 3$ biological independent experiments are presented. **g**, **h** Representative images and quantitative analyses of expression of proliferation marker Ki67 detected by immunofluorescence (green) in control hPASMCs. DNA was visualized using 4′,6′-diamidino-2-phenylindole (DAPI; blue). Data from $n = 4$ biological independent experiments are presented. Scale bar: 50 μm. **i** Real-time quantitative PCR analyses of *AXL* expression normalized to the glyceraldehyde-3-phosphate dehydrogenase (*GAPDH*) house-keeping gene in mRNA from hPASMCs after transduction with lentiviral constructs coding for Axl overexpression and Axl knock-down (short hairpin RNA), compared with control-infected cells. Data from $n = 4$ biological independent experiments are presented. **j**, **k** Graph representing quantification of BrdU incorporation in hPASMCs after PDGF-BB stimulation, pre-treated with increasing concentrations of R428, **j** after lentiviral overexpression of Axl and control construct, and **k** after lentiviral transduction of Axl-silencing short hairpin RNA and control construct. Percentage of proliferated cells was normalized to untreated control cells. Data from $n = 2$ biological independent experiments in triplicate are presented. Statistical analysis was performed using one-way analysis of variance (ANOVA) with Tukey's post hoc test for multiple comparisons. **a–h** $^{**}P < 0.01$, $^{***}P < 0.001$, and $^{****}P < 0.0001$ versus DMSO-treated control cells; $^{#}P < 0.05$, $^{##}P < 0.01$, $^{###}P < 0.001$, and $^{####}P < 0.0001$ versus PDGF-BB-treated cells. Open, green, and gray circles define control (co), PDGF-BB-, and PDGF-BB + R428-treated conditions, respectively. **i** $^{****}P < 0.0001$ versus control-infected (co and co sh) hPASMCs. All the data are presented as mean ± SEM.

**R428 targets apoptosis, inflammation, and autoimmunity in experimental PH**. To understand the mechanism underlying the detrimental effect of Axl inhibition in experimental PH, several key processes associated with the pathogenesis of PAH and Axl signaling[26–29] were investigated. R428 increased Caspase-3 activity in SuHOX and MCT rat lung tissue (Fig. 4a, b and Supplementary Fig. 4a, b) and SuHOX pulmonary vessel ECs (Fig. 4c). A robust increase in p53 protein and its downstream target p21 was noted after R428 treatment in both SuHOX and MCT rat lungs (Fig. 4a, d, e and Supplementary Fig. 4a, c, d). R428 also increased phosphatidylserine levels and anti-cardiolipin

IgG in plasma of SuHOX rats compared with vehicle-treated and NOX controls, respectively, indicating the activation of apoptotic signals (Fig. 4f, g). The inflammatory markers nitrotyrosine and phospho-signal transducer and activator of transcription 3 (phospho-STAT3) and mRNAs of interferon gamma (*IFN-γ*) and interleukin-6 (IL-6) were increased in the lungs of rats receiving R428 compared with the vehicle-treated control rats (Fig. 4h, i, k, l and Supplementary Fig. 5a, c, d). Furthermore, the expression of IFN-γ-induced neopterin, a molecule, which augments immune response during inflammation, was elevated in the circulatory system of R428-treated rats (Fig. 4m). Both rat models exhibited

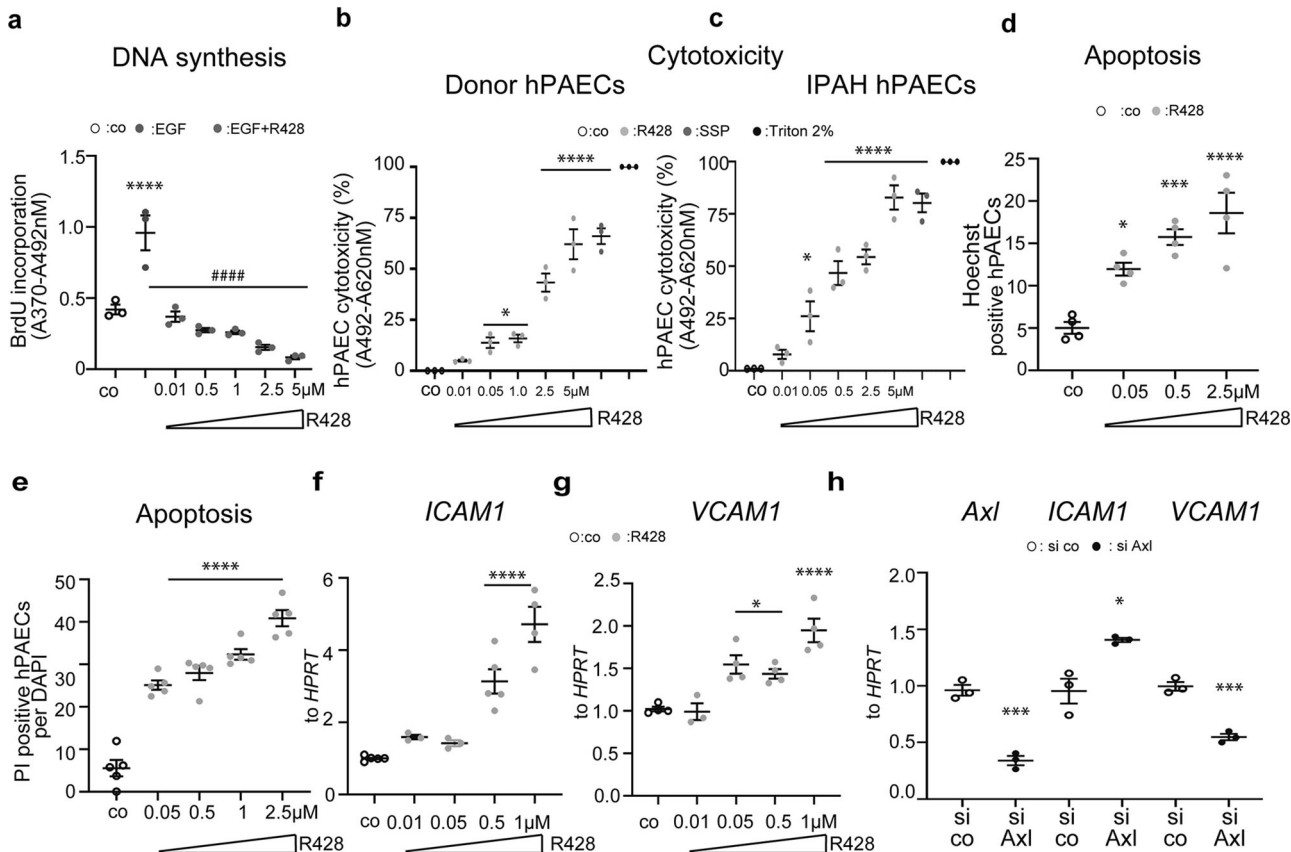

**Fig. 2 Axl inhibition suppresses the proliferation and induces the cytotoxicity and apoptosis of human pulmonary arterial endothelial cells (hPAECs). a** HPAECs from healthy individuals exposed to 10 ng/ml of epidermal growth factor (EGF) with or without R428 at indicated concentrations. Control cells (co) were exposed to dimethyl sulfoxide (DMSO). Data from $n = 3$ biological independent experiments performed in triplicate are presented. Open, green, and dark gray circles define untreated control (co), EGF, and EGF + R428 conditions, respectively. **b, c** Cytotoxicity in donor and IPAH hPAECs was determined by detecting lactate dehydrogenase (LDH) release into the cell culture media [$A_{492nm}$–$A_{620nm}$]. Cells treated with Staurosporine (SSP) were used as a positive control. Data are reported as percentage release of LDH compared to DMSO-treated control (co) cells. Data from $n = 3$ biological independent experiments performed in triplicates are presented as mean ± SEM. Number of **d** Hoechst and propidium iodide (PI) **e** positive hPAECs after R428 treatment. Data from $n = 4$ (Hoechst) and $n = 5$ (PI) independent experiments presented as the $n$-fold change normalized to dimethyl sulfoxide (DMSO)-treated control cells (co). **f, g** Relative mRNA expression of **f** intercellular adhesion molecule 1 (*ICAM1*) and **g** vascular cell adhesion molecule 1 (*VCAM1*) normalized to hypoxanthine guanine phosphoribosyl transferase (*HPRT*) as a reference gene after R428 treatment. Data from $n = 3$-5 biological independent experiments are presented as the $n$-fold change ($2^{-\Delta\Delta Ct}$) compared with DMSO-treated control (co). *P* values for distinct conditions are given for their comparison with DMSO-treated control cells (co). Open and gray circles define DMSO- and R428-treated conditions, respectively. **h** Analysis of *AXL*, *ICAM1*, and *VCAM1* after small interfering RNA (siRNA) knock-down of *AXL* in hPAECs. *AXL*, *ICAM1*, and *VCAM1* mRNA expression were normalized to *HPRT* as a reference gene. Data from $n = 3$ biological independent experiments in triplicate are presented as the $n$-fold change ($2^{-\Delta\Delta Ct}$) compared with siRNA scrambled control (si co). Opened and black circles define si co and siAXL conditions, respectively. For statistical analysis in **a–g** one-way ANOVA with Tukey's post hoc test for multiple comparisons $^{*}P < 0.05$, $^{***}P < 0.001$, and $^{****}P < 0.0001$ versus DMSO-treated control cells; $^{####}P < 0.0001$ versus EGF-treated cells were implemented; for **h** Student's *t*-test $^{*}P < 0.05$ and $^{***}P < 0.001$ versus scrambled siRNA control-transfected cells. All the data are presented as mean ± SEM.

an increase in the number of CD68 perivascular cells upon R428 administration (Supplementary Fig. 5e–h). R428 had a slight, though not significant impact on suppressor of cytokine signaling 3 (SOCS3) expression (Fig. 4j and Supplementary Fig. 5a) and no influence on p65 phosphorylation (Supplementary Fig. 5a, b). Previous studies demonstrated that Axl inhibition induces an adaptive immune resistance evidenced by unregulated programmed death ligand 1 (PD-L1) expression[29]. Interestingly, expression of the immune checkpoint programmed death-1 (PD-1)/programmed death ligand 1 (PD-L1) pathway, which plays a key role in modulating immune response, was markedly induced in experimental rat lungs (Supplementary Fig. 6a–e). R428 markedly enhanced the PD-1/PD-L1 axis compared with vehicle in lungs from both rat models (Supplementary Fig. 6a–e), and increased mRNA levels of C-X-C motif chemokine ligands

*CXCL10* and *CXCL11* (encoding chemokines associated with type-1 T cell recruitment and functionality) (Supplementary Fig. 6f, g). Taken together, our data indicate that Axl inhibition augments both apoptotic and inflammatory responses in experimental PAH.

**R428 inhibits BMPR2 signaling in experimental PH.** SuHOX rats receiving R428 exhibited remarkable attenuation of the BMPR2 signaling pathway, as determined by a decrease of *BMPR2* and inhibitor of DNA binding 1 (*ID1*) and *ID2* on mRNA, and BMPR2 and ID1 on protein levels compared with control rats (Fig. 5a–f). Furthermore, R428 decreased circulating E-selectin (a downstream target of BMPR2 in ECs) compared with vehicle-treated control rats (Fig. 5d). A concomitant decline

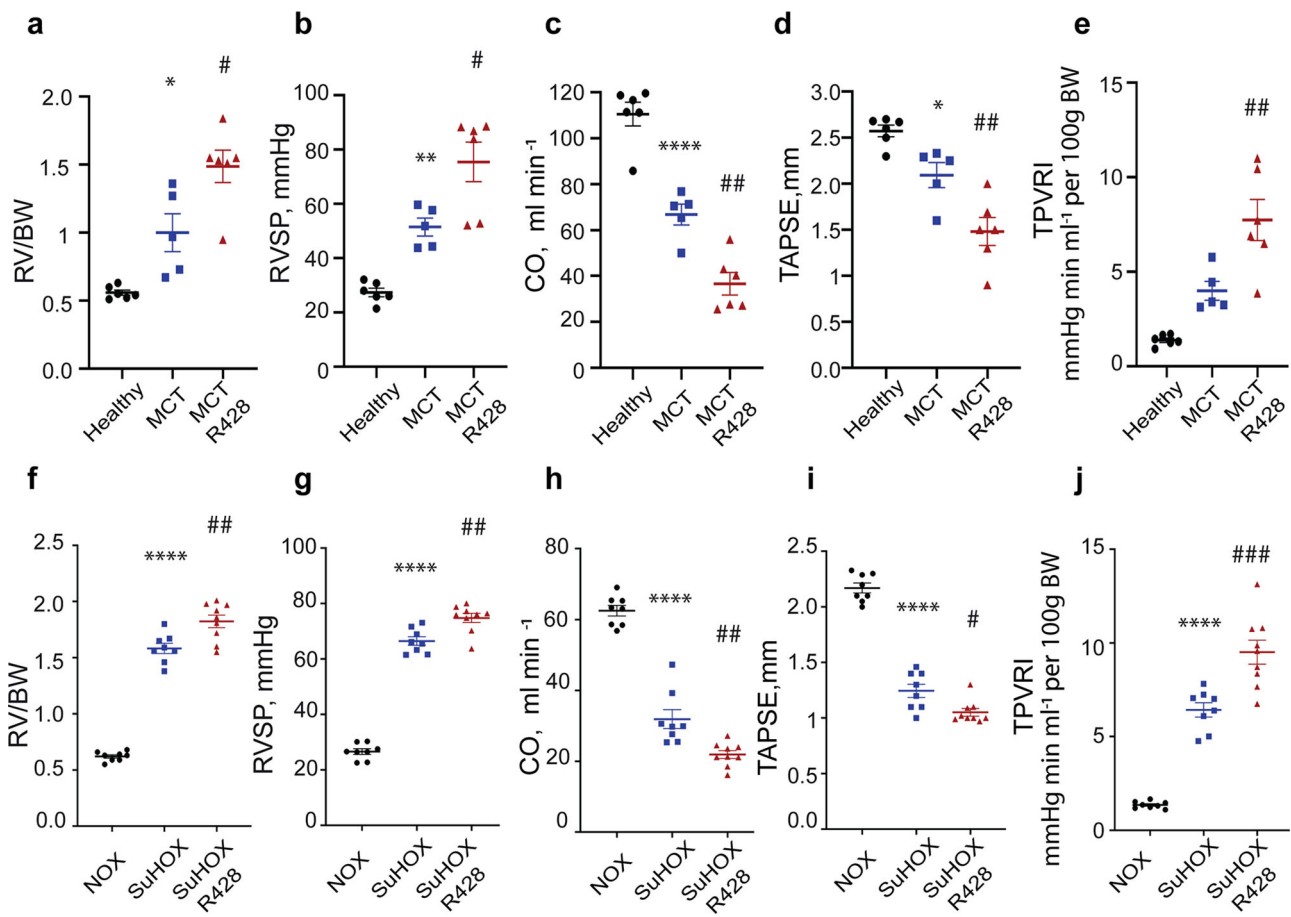

**Fig. 3 Axl inhibition aggravates hemodynamics, right ventricle (RV) hypertrophy, and RV function in monocrotaline (MCT) and Sugen 5416/hypoxia (SuHOX) experimental rat models of pulmonary hypertension. a, f** RV hypertrophy, measured as the ratio of RV to body weight (RV/BW). **b, g** Right ventricular systolic pressure (RVSP, mmHg). **c, h** Cardiac output (CO, ml/min). **d, i** Tricuspid annular plane systolic excursion (TAPSE, mm). **e, j** Total pulmonary vascular resistance index (TPVRI, mmHg/min/ml per 100 g body weight). **a–e** Echocardiography measurements were performed in male Sprague-Dawley rats at baseline, 21 days after MCT administration, and again on day 35 after 2 weeks of daily treatment with R428 (100 mg/kg body weight). Hemodynamics and cardiac function were assessed on day 35 (MCT + R428, $n = 6$; MCT + placebo [MCT], $n = 5$; healthy, $n = 6$). **f–j** Echocardiography measurements were performed in male Wistar-Kyoto rats at baseline, 21 days after Sugen 5416 administration and subsequent 3-week hypoxic period, and on day 35 after 2 weeks of daily treatment with R428 (100 mg/kg body weight) under normoxia. Hemodynamics and cardiac function were assessed on day 35 (SuHOX + R428, $n = 9$; SuHOX + placebo [SuHOX], $n = 8$; normoxic rats [NOX], $n = 8$). Black circles NOX and Healthy controls; blue squares MCT and SUHOX, and red triangles MCT + R428 and SuHOX + R428. All statistical analyses were performed using one-way ANOVA with Tukey's post hoc test for multiple comparisons. $^{*}P < 0.05$, $^{**}P < 0.01$, and $^{****}P < 0.0001$ for placebo-treated rats versus respective controls; $^{\#}P < 0.05$, $^{\#\#}P < 0.01$, and $^{\#\#\#}P < 0.001$ for R428-treated rats versus placebo-treated controls. All the data are presented as mean ± SEM.

in SMAD1/5/8 phosphorylation was observed in R428-treated rats (Fig. 5e, f). The expression of phosphorylated SMAD2/3 proteins was markedly enhanced upon R428 treatment, indicating that Axl inhibition on PAH is linked with a switch between TGF-β-SMAD2/3 and BMPR2-SMAD1/5/8 signaling (Fig. 5e, f). In SuHOX lungs, Axl protein correlated positively with BMPR2 abundance (Fig. 5g). Similarly, a highly significant correlation between Axl and pSMAD1/5/8, but not pSMAD2/3 was found in the lungs of the SuHOX rats (Fig. 5g). Outstandingly, western blot analyses of lung protein extracts indicated that Axl inhibition does not impact the expression of Activin A receptor type 2a (ActR-2a) and Activin A receptor type 2B (ActR-2b) (Supplementary Fig. 7a, b). These data indicate that Axl may contribute to BMPR2 signaling.

**R428 promotes vascular remodeling and pulmonary EC apoptosis.** We next examined whether Axl inhibition modulates vascular remodeling in MCT and SuHOX treated rats. R428 increased neointima thickness of small pulmonary arteries in the

MCT rat model (Fig. 6a–c). Interestingly, R428 reduced the neointima layer in the SuHOX model (Fig. 6f–h). Furthermore, R428 increased the number of fully occluded vessels (Fig. 6d, i). Vascular rarefaction is an important contributor to the hemodynamic changes in patients with PAH[30], which may occur by EC apoptosis in distal arteries, evidenced by increased numbers of circulating ECs[31]. Axl inhibition resulted in amelioration in total number of small vessels in both rat models of experimental PAH (Fig. 6e, j). R428 increased the percentage of apoptotic PAECs, but not PASMCs in distal pulmonary arteries (Fig. 6k–n), and declined the number of both pulmonary microvascular endothelial cells (PMVECs) (Fig. 6o, p) and vWF-positive intrapulmonary ECs, compared with the vehicle (Fig. 6q). PCNA immunostaining analyses reveal no remarkable effect on vascular cell proliferation (Supplementary Fig. 7b–e).

**Modulation of BMPR2 activity by Axl in hPAECs.** R428 significantly and dose-dependently reduced BMPR2 protein in hPAECs (Fig. 7a, b) and *BMPR2* transcripts in hPASMCs

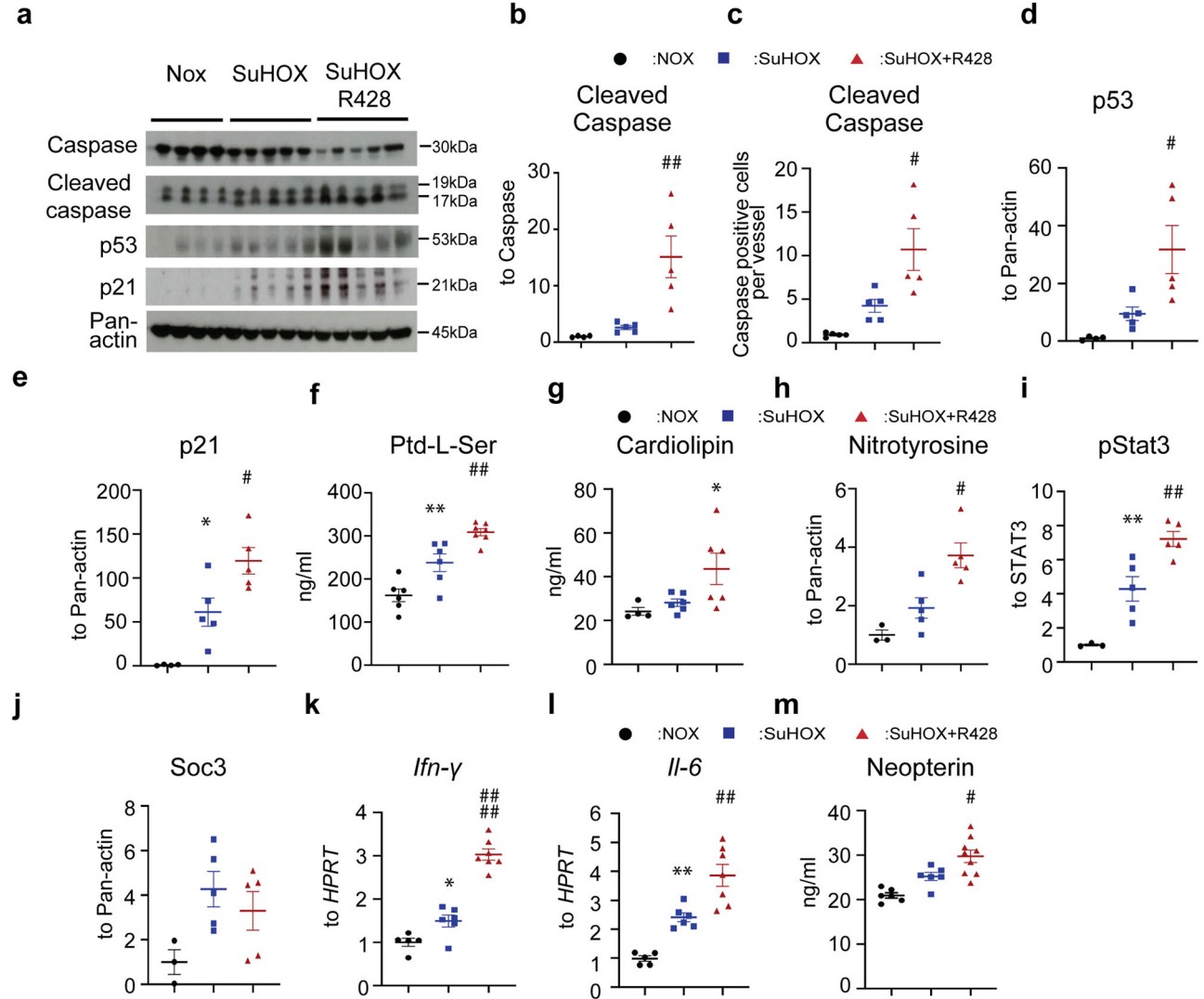

**Fig. 4 Effect of R428 on apoptotic, inflammatory, and immune responses in the Sugen 5416/hypoxia (SuHOX) experimental rat model of pulmonary hypertension. a** Representative western blots of markers of apoptosis. **b** Subsequent densitometric quantification of Cleaved Caspase. **c** Quantitative analyses of Caspase-positive cells per vessel. **d, e** Densitometric quantification of **d** p53 and **e** p21. **f, g, m** Enzyme-linked immunosorbent assay (ELISA) for analysis of the presence of phosphatidylserine (Ptd-L-Ser), anti-cardiolipin antibody IgG (ACA-IgG) and neopterin in plasma of rats. **h–j** Densitometric quantification of markers of inflammation **h** nitrotyrosine, **i** phosphorylated signal transducer and activator of transcription 3 [phospho-STAT3], and **j** suppressor of cytokine signaling 3 [SOCS3]) in total lung homogenates. **k, l** Real-time quantitative PCR analysis of **k** interferon γ (*Ifn-γ*) and **l** interleukin 6 (*Il-6*) mRNA expression in lungs of rats. Data are presented as the *n*-fold change ($2^{-\Delta\Delta Ct}$) normalized to hypoxanthine guanine phosphoribosyl transferase (*Hprt*) as a reference gene, compared with NOX. The numbers of biological independent rats included in the analyses were as follows. Western blots and densitometric quantification: normoxic rats (NOX), $n = 4$; SuHOX + placebo (SuHOX), $n = 5$; SuHOX + R428, $n = 5$. Real-time quantitative PCR: NOX, $n = 5$; SuHOX, $n = 6$; SuHOX + R428, $n = 7$. ELISAs: NOX, $n = 4$–6; SuHOX, $n = 6$; SuHOX + R428, $n = 6$–9. Black circles NOX controls; blue squares SUHOX; red triangles SuHOX + R428. Statistical analysis was performed using one-way ANOVA with Tukey's post hoc test for multiple comparisons. $^*P < 0.05$ and $^{**}P < 0.01$ versus NOX; $^\#P < 0.05$, $^{\#\#}P < 0.01$, and $^{\#\#\#\#}P < 0.0001$ for SuHOX + R428 versus SuHOX. All the data represent the mean ± SEM.

(Supplementary Fig. 8a). In hPAECs, the decline of BMPR2 was associated with a decrease in mRNA levels of its downstream target, E-selectin (Fig. 7c), indicating that Axl may contribute to BMP responsiveness in hPAECs. BMP signaling is majorly controlled by BMP2, 4, 6, 9, and 10 factors in vascular endothelium[32]. Axl phosphorylation was found to occur upon short-term stimulation with all studied BMPs except BMP2 (Fig. 7d), indicating a selective involvement of distinct BMPs in Axl activation. Of the studied BMPs, BMP9 has been shown previously to promote PAEC survival[33] and protect PMVECs from apoptosis[34]. We therefore investigated the effect of BMP9 on Axl cascade activation. Short-term BMP9 stimulation resulted in a significant and time-dependent phosphorylation of Axl (Fig. 7e), while Gas6

enhanced the activation of BMPR2 signaling via phosphorylation of SMAD1/5/8 and ID1 increase (Fig. 7f–h and Fig. 8a, b). Importantly, Gas6 neutralizing antibody completely obstructed Gas6-mediated SMAD1/5/8 activation (Supplementary Fig. 8b) and BMPR2 increase (Supplementary Fig. 8c). Inhibition of BMP signaling by LDN-193189 diminished the Gas6-induced phosphorylation of SMAD1/5/8 and *ID1/ID2* increase (Fig. 7g, h). Both ligands protected hPAECs from R428-induced apoptosis (Fig. 7i, j). Moreover, Axl blockage by R428 significantly inhibited BMP9-induced BMP signaling at the expense of ID1 protein decrease (Fig. 8a, b). The knock-down of Axl and BMPR2 each diminished both BMP9- and Gas6-induced SMAD1/5/8 activation directly and reciprocally (Fig. 8c, d).

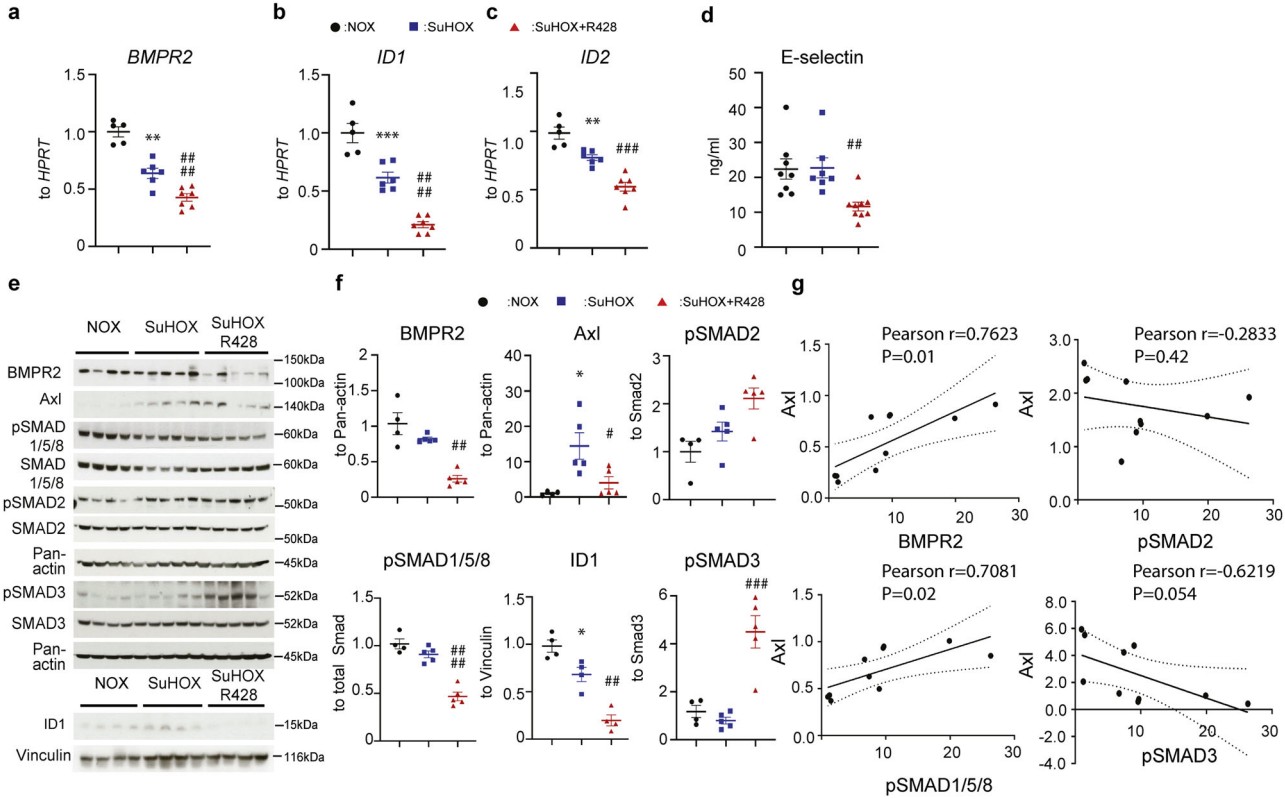

**Fig. 5 Effect of R428 on BMPR2 signaling pathway in the Sugen 5416/hypoxia (SuHOX) experimental rat model of pulmonary hypertension. a–c** Real-time quantitative PCR analysis of bone morphogenetic protein receptor 2 (*BMPR2*) and inhibitor of DNA binding 1 (*ID1*) and 2 (*ID2*) mRNA expression. Data are presented as the *n*-fold change ($2^{-\Delta\Delta Ct}$) normalized to hypoxanthine guanine phosphoribosyl transferase (*Hprt*) as a reference gene, compared with normoxic control rats (normoxic [NOX], $n = 5$; SuHOX + placebo [SuHOX], $n = 6$; SuHOX + R428, $n = 7$ biological independent animals). **d** Enzyme-linked immunosorbent assays for analysis of the presence of E-selectin in plasma of biological independent rats (NOX, $n = 8$; SuHOX, $n = 7$; SuHOX + R428, $n = 9$ biological independent animals). **e** Western blots and **f** subsequent densitometric quantification of BMPR2, Axl, phospho-SMAD1/5/8, phospho-SMAD2, phospho-SMAD3, and ID1 expression in lung homogenates of representative samples from all three studied groups (NOX, $n = 4$; SuHOX, $n = 4$-5; SuHOX + R428, $n = 4$-5 biological independent animals). Vinculin and Pan-actin served as a loading controls. **g** Analyses of the correlation between BMPR2, pSMAD1/5/8, pSMAD2, pSMAD3, and Axl expression in total lungs of rats from SuHOX experimental model. Pearson correlation coefficient (*r* value) and the significance (two-tailed *P* value) of each correlation analysis are represented. Black circles NOX controls; blue squares SUHOX; red triangles SuHOX + R428. Statistical analysis was performed using one-way ANOVA with Tukey's post hoc test for multiple comparisons. $^{*}P < 0.05$, $^{**}P < 0.01$, and $^{***}P < 0.001$ for SuHOX versus NOX; $^{\#}P < 0.05$, $^{\#\#}P < 0.01$, $^{\#\#\#}P < 0.001$, and $^{\#\#\#\#}P < 0.0001$ for SuHOX + R428 versus SuHOX. All the data represent the mean ± SEM.

A significant reduction of *ID1*, and *ID2* mRNA transcripts was also noted in Axl small interfering RNA-treated hPAECs, indicating that Axl is required for BMP signaling transduction to Smad1/5/8 phosphorylation and ID1/ID2 increase in ECs (Fig. 8e). Similarly, BMPR knock-down lessened Gas6-prompted augmentation of *ID1/ID2* expression, suggesting an interaction of both pathways.

**Axl is physically associated with BMPR2.** Next, we looked at the expression of Axl receptor in clinical PAH. Interestingly, despite an elevation of Axl expression in lung homogenates of clinical PAH (Supplementary Fig. 8d), no changes in the expression profile of Axl protein were found in hPASMCs isolated from non-PAH individuals and IPAH patients (Supplementary Fig. 8e). Outstandingly, Axl demonstrates a highly significant positive correlation with BMPR2 abundance in hPASMCs (Supplementary Fig. 8g). Concomitantly, a substantial decline in *AXL* mRNA in MDVs of patients with IPAH compared with non-PAH controls was determined (Fig. 9a). Similarly, BOECs isolated from patients with IPAH exhibited a tendency of reduced *AXL* expression (Fig. 9b). Moreover, hPAECs from IPAH patients exhibited a decline of Axl protein in comparison to non-PAH

control PAECs (Fig. 9c, d). A robust correlation between Axl and BMPR2 protein expression in hPAECs was found (Fig. 9e). A decay of Axl protein in cultured hPAECs was corresponding to its drop in pulmonary endothelium in lung specimens of patients with IPAH, as revealed by immunofluorescence staining with von Willebrand factor (vWF) and α-smooth muscle actin (α-SMA) antibodies (Fig. 9f and Supplementary Fig. 8h). In HEK293 cells, overexpressed Axl co-immunoprecipitated with overexpressed BMPR2 and vice versa (Fig. 9g, h). In hPAECs, endogenous BMPR2 co-immunoprecipitated with endogenous Axl, indicating that Axl is a novel interacting partner of BMPR2 (Fig. 9i). Both immunoprecipitations and proximity ligation assays exhibited enhanced receptor–receptor interaction after simultaneous BMP9/Gas6 stimulations of hPAECs (Fig. 9j–l). R428 efficiently blocked receptor–receptor interaction under basal and stimulated conditions (Fig. 9k, l). Taken together out data point to a previously unrecognized association between RTK Axl and BMPR2 in PAECs.

## Discussion
To the best of our knowledge our study is the first investigation of the role of the Gas6/Axl pathway in PH. We found an altered Axl

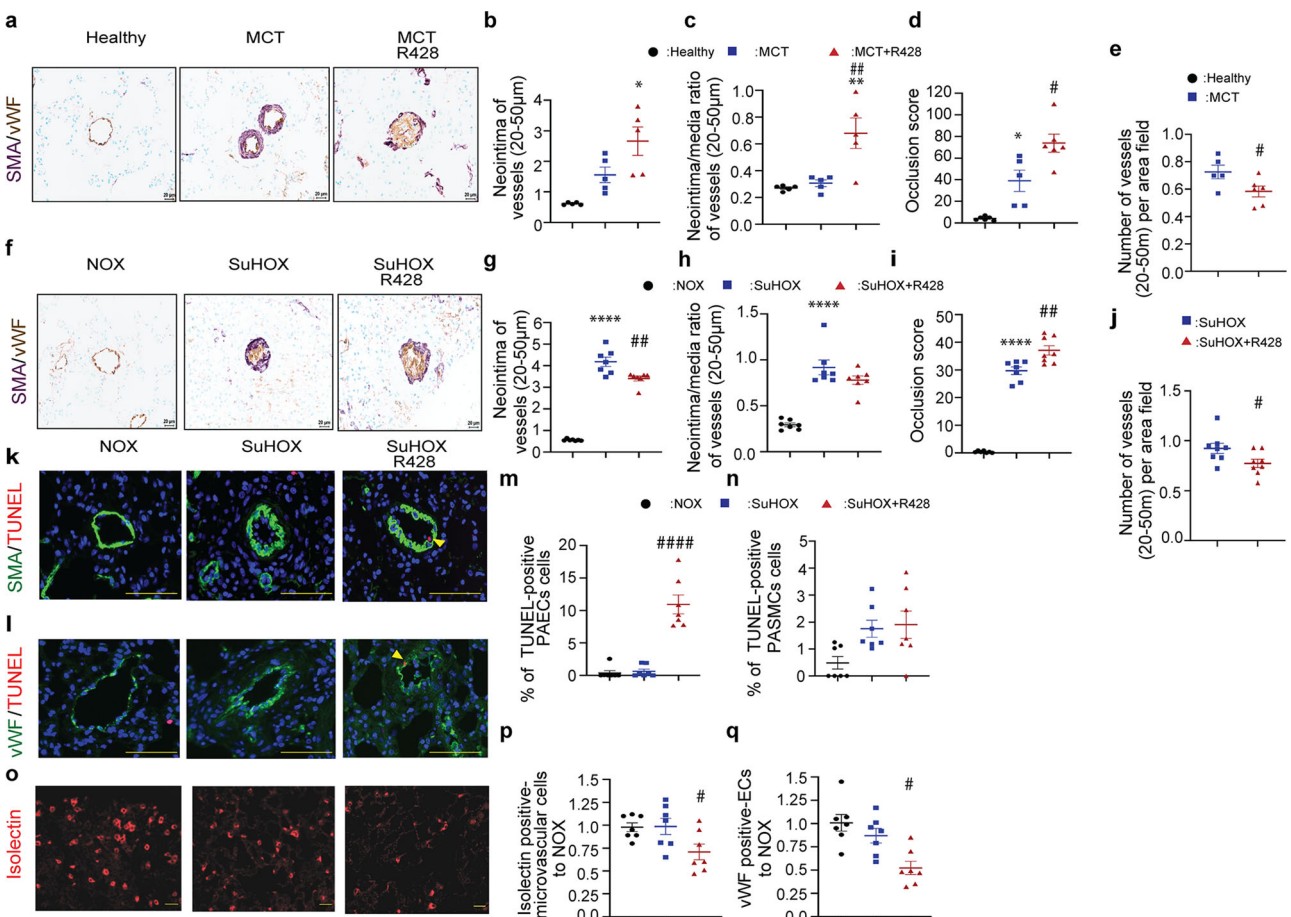

**Fig. 6 Axl blockage by R428 administration exacerbates lung tissue remodeling in monocrotaline (MCT) and Sugen 5416/hypoxia (SuHOX) experimental rat models of pulmonary hypertension. a, f** Representative images of immunohistological staining of lung sections for von Willebrand factor (vWF; brown) and α-smooth muscle actin (α-SMA; violet) in **a** monocrotaline (MCT) and **f** Sugen 5416/hypoxia (SuHOX) models of pulmonary hypertension. Scale bar: 20 μm. **b, g** The neointima thickness (μm) and **c, h** the neointima/media of the studied groups determined by Elastica-van-Gieson staining in the vessels with a diameter of 20–50 μm. **d, i** Number of occluded vessels with a diameter of 20–50 μm (per 100 vessels). **e, j** Number of vessels counted per lung (normalized per surface area) **a–i** MCT model: healthy, $n = 5–6$; MCT + placebo [MCT], $n = 5$; MCT + R428, $n = 6$. SuHOX model: NOX, $n = 7$; SuHOX + placebo [SuHOX], $n = 7–8$; SuHOX + R428, $n = 7–8$ biological independent animals. Data are presented as mean ± SEM of all counted vessels per lung. **k, l** Representative images and **m, n** quantification of TUNEL-positive cells in lungs of SuHOX rats and controls (**k** α-SMA [green] and **l** vWF [green]; nuclei, blue (diamidino-2-phenylindole [DAPI]). Graphs represent the percentage of PASMCs and PAECs positive for TUNEL in distal pulmonary vessels of lung rats. **o** Representative images and **p** quantitative analysis of immunofluorescent staining of lung sections for pulmonary microvascular endothelial cells (PMVECs) using isolectin IB4 (red). Scale bar = 50 μm. **q** Quantitative analysis of immunofluorescent staining of lung sections for pulmonary endothelial cells (PAECs) using vWF antibody. **m–q** NOX, $n = 7$; SuHOX + placebo [SuHOX], $n = 7$; SuHOX + R428, $n = 7$ biological independent animals. **b–d** $^{*}P < 0.05$, $^{**}P < 0.01$ for MCT (or MCT + R428) versus healthy; **g–i** $^{****}P < 0.0001$ for SuHOX (or SuHOX + R428) versus NOX; **c–j** $^{#}P < 0.05$, $^{##}P < 0.001$, and $^{####}P < 0.001$ for MCT + R428 and SuHOX + R428 versus MCT or SuHOX, respectively. Black circles NOX controls; blue squares SUHOX; red triangles SuHOX + R428. **b–q** Statistical analyses were performed using one-way ANOVA with Tukey's post hoc test for multiple comparisons. **e, j** Student's $t$-test was applied. All the data represent the mean ± SEM.

expression in experimental PH and patients with IPAH and showed that Axl inhibition by R428 worsened experimental PH and augmented hPAEC apoptosis. Both MCT and SuHOX animal models have been employed for decades to develop experimental PAH, by inducing both PH and RVH. MCT, a highly toxic alkaloid, when ingested stimulates widespread pneumotoxicity, connected with increased cell proliferation and systemic inflammation[35]. In SuHOX model, SU5416-mediated VEGFR2 blockade, in combination with hypobaric hypoxia, induces a severe PH with elements of inflammation and angio-obliteration[36]. Both models are characterized by EC injury that also have been associated with BMPR2 dysregulation and BMP signaling deficiency. In contrast to SuHOX model of severe PAH, MCT model does not exhibit EC-mediated angio-obliteration, but reveals pronounced pulmonary arterial hypertrophy[35]. In our

study, Axl blockage of MCT-administered rats resulted in a significant increase in neointima/media ratio, suggesting that EC dysfunction in this model is a direct consequence of EC damage.

These data are consistent with previously published results in which Axl activation rescued human umbilical vein ECs and hPAECs from apoptosis[37–39], and systemic deletion of Axl in mice increased apoptosis in vein graft remodeling[40]. Moreover, the Gas6/Axl pathway has been shown to contribute to high-glucose-induced EC dysfunction[41], suggesting that Gas6-dependent signaling may be relevant to EC survival in the quiescent environment of the vessel wall. Interestingly, while Axl expression is markedly induced in whole-lung tissue samples of IPAH patients, in comparison to non-PAH control individuals, no difference of the Axl expression in hPASMCs from IPAH patients versus non-PAH control cells was noted. In contrast, the

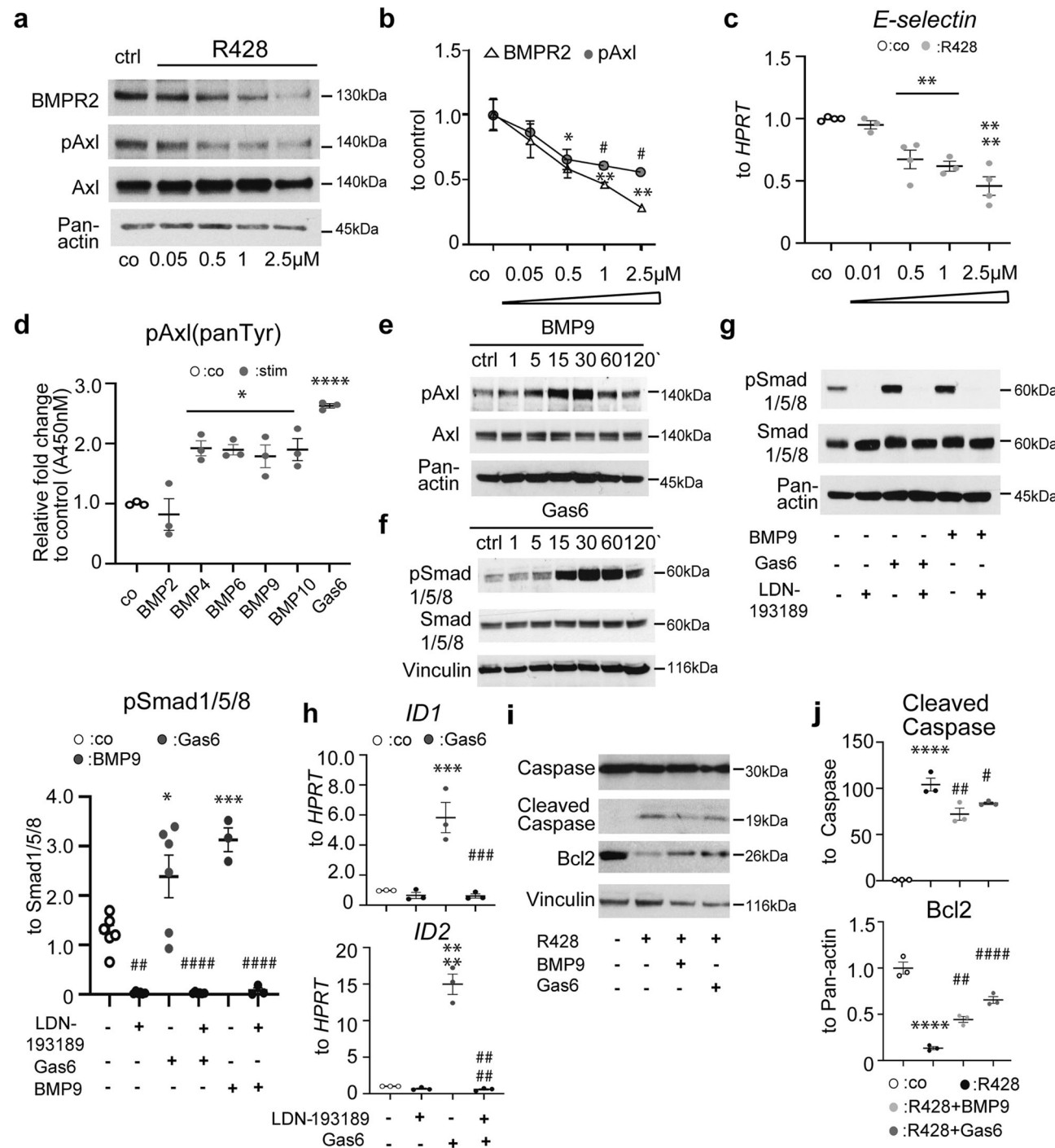

expression of Axl was strongly induced in rat and mouse PH-PASMCs. Outstandingly, Axl receptor was prominently reduced both in the pulmonary endothelium and isolated hPAECs of patients with IPAH, indicating a selective and EC-specific mode of regulation. We found a cross-talk between the Gas6/Axl and BMP9/BMPR2 signaling pathways and identified a previously unrecognized physical association between Axl and BMPR2. Interactions of Axl with other receptors of TAM and non-TAM families have been implicated in various functional outcomes. The association of Axl and integrin complexes has been postulated to contribute to Axl-mediated anti-apoptotic effects induced by laminar shear stress in ECs[14], while the interaction of Axl with HER2 at the cell surface is required for invasion and metastasis in HER2-positive breast cancer[42]. Axl is present in local clusters

together with ErbB and MET receptors on the plasma membrane of cancer cells[11]. Such clustering of plasma membrane receptors results in activation-dependent enhancement of interactions after ligand stimulation[43]. In our experiments, an interaction between Axl and BMPR2 was facilitated by concomitant ligand stimulation, signifying receptor clustering at the plasma membrane of ECs. This association was weakened by the addition of R428 in hPAECs. BMPR2 presence in pulmonary arteries is a critical determinant of PAH, as patients with IPAH without a BMPR2 mutation or with PAH associated with other related medical conditions exhibit a decay in the expression of BMPR2 in pulmonary arteries[44]. The interaction of Axl with BMPR2 at the cell surface of PAECs might be required for maintaining the local clusters between receptors and thereby EC homeostasis, while the

**Fig. 7 Mutually mediated interaction of bone morphogenetic receptor 2 (BMPR2) and Axl signaling pathways in human pulmonary arterial endothelial cells (hPAECs). a** Western blots and **b** subsequent densitometry quantification of BMPR2 and phospho-Axl in hPAECs exposed to R428 at the indicated concentrations. Opened triangles and dark gray circles denote BMPR2 and pAxl, respectively. **c** Relative mRNA expression of *E-selectin* normalized to hypoxanthine guanine phosphoribosyl transferase (*HPRT*) as a reference gene. Data from $n = 4$ biological independent experiments are presented as the $n$-fold change ($2^{-\Delta\Delta Ct}$) compared with DMSO-treated control cells. **d** Endogenous levels of tyrosine-phosphorylated Axl analyzed by PathScan® phospho-Axl (panTyr) Sandwich ELISA. **c, d** Opened circles define untreated control (co) cells, light gray circles define R428, dark gray circles define BMP2-10 and Gas6 conditions. **e, f** Western blots of **e** Axl, and **f** SMAD1/5/8 phosphorylation after BMP9 (10 ng/ml) and Gas6 (200 mg/ml) stimulation. **g** Western blots and subsequent densitometry quantification of SMAD1/5/8 phosphorylation after LDN-193189 (2.5 μM) treatment followed by Gas6 and BMP9 stimulations (30 min). **h** Relative mRNA expression of inhibitor of DNA binding 1 and 2 (*ID1, ID2*) normalized to *HPRT* as a reference gene. Data from $n = 4$ biological independent experiments are presented as the $n$-fold change ($2^{-\Delta\Delta Ct}$) compared with control-treated control cells. **g, h** Untreated control (co) is defined with opened circles. Green, purple and black circles define Gas6-, BMP9-, and LDN19189-treated conditions, respectively. **i, j** Western blots and subsequent densitometry quantification of Cleaved Caspase and Bcl2 after R428 treatment followed by Gas6 or BMP9 (24 h). **b, d, g–j** Data from $n = 3$ biological independent experiments are presented as the $n$-fold change compared with **b, i, j** DMSO-treated or **d, g, h** untreated control (co) cells. Untreated control (co) is defined with opened circles. Black, light gray, and dark circles define R428-, R428 + BMP9-, and R428 + Gas6-treated conditions, respectively. Statistical analyses were performed using one-way ANOVA with Tukey's or Newman–Keuls post hoc test for multiple comparisons. **a–j** $^*P < 0.05$, $^{**}P < 0.01$, $^{***}P < 0.001$, $^{****}P < 0.0001$ versus untreated or DMSO-treated control cells; **g, h** $^{##}P < 0.01$, $^{###}P < 0.001$, and $^{####}P < 0.0001$ for LDN-193189 versus BMP and Gas6 stimulations, and **i, j** $^{#}P < 0.05$, $^{##}P < 0.01$, and $^{####}P < 0.0001$ R428 + BMP9 and R428 + Gas6 versus R428. In **a, e, g** Pan-actin, in **f, i** Vinculin served as a loading control. All the data represent the mean ± SEM.

inhibition of Axl signaling might trigger EC apoptosis, eventually enhancing dysfunctional BMPR2 signaling in the context of PAH. Like the others, we did not detect the changes of BMPR2 and its downstream signaling component phopsho-Smad1/5/8 in the whole-lung lysates of rats of SuHOX model[45], however Axl inhibition triggered the decline of BMPR2 signaling both on the level of BMPR2 and phospho-Smad1/5/8 proteins, as well as EC-specific downstream target of BMPR2, E-selectin, confirming the detrimental impact of Axl inhibition in the context of PAH. Likewise, the inhibition of Axl in cultured hPAECs significantly inhibited BMPR2 signaling at the expense of decrease of Smad1/5/8 phosphorylation and ID1, ID2, and E-selectin expression, verifying that Axl is required for BMPR2 signaling in ECs. Interestingly, Gas6-mediated Axl activation enhanced BMPR2 signaling and protects from R428-induced hPAEC apoptosis, while inhibition of BMPR2 signaling by LDN-193189 diminished the Gas6-induced phosphorylation of SMAD1/5/8 and ID1 increase, suggesting that Gas6/Axl might play a protective role in the pulmonary endothelium in the context of PAH.

Loss of precapillary vessels and impaired vascular regeneration are two sides of vessel remodeling in PAH[46]. Gas6 (via Axl activation) may, similarly to BMP4[47] and BMP9[33], enhance PAEC survival and potentially increase tube formation, both essential for the recovery of remodeled vessels in PAH. Furthermore, Axl blockage may prolong the initial period of EC predisposition to apoptosis in SuHOX rats, which markedly affects a secondary hyper-proliferative EC phenotype culminating in PH. In this regard, the sustained levels of Axl phosphorylation may serve as an essential prerequisite for EC survival in PAH. Importantly, BMP9-induced Axl phosphorylation, confirms the protective role of Axl phosphorylation in cultured hPAECs.

TAM RTKs have been shown to reduce JAK/STAT-mediated immune responses[21,48], while TAM triple knock-out activates the production of inflammatory cytokines[49]. Blockage of Axl in our disease settings induced pro-inflammatory signaling, consistent with a recent study in which R428 antagonized the anti-inflammatory effect of Gas6 in ischemia reperfusion-induced acute lung injury, suggesting that Gas6-induced Axl phosphorylation is protective not only in alveolar epithelium[50], but also in the vascular endothelium.

Recently, Axl was demonstrated to be unsusceptible to combined irradiation and PD-L1 immunotherapy, while Axl knockdown contributed to reprogramming of the immunological microenvironment, suggesting that disabling Axl signaling may promote adaptive immunity[51,52]. The binding of PD-L1 to PD-1

transmits an inhibitory signal suppressing adaptive immune responses. As expected, Axl inhibition in our experimental models of PH strongly activated PD-1/PD-L1 signaling, indicating downregulation of the adaptive immune system. It was previously postulated that antibodies against the vascular endothelium might promote endothelial apoptosis, which might serve as a hallmark of autoimmune disorders[53]. Moreover, Axl inhibition was shown to upregulate neopterin, a marker of PAH and inoperable chronic thromboembolic PH[54] which can be released by both inflammatory cells and ECs[55]. In R428-treated rats, the neopterin increase was accompanied by a remarkable upregulation of anti-phospholipid antibodies that bind and activate ECs, resulting in their apoptosis[56,57]. Furthermore, Axl inhibition enhanced cardiolipin antibody, a molecule essential for the recruitment of apoptotic factors and inflammasome formation[58]. Thus, we assume that Axl inhibition results in autoantibody production and associated vascular endothelial injury, which contributes, in combination with other genetic and environmental factors, to the emergence of PAH. Gas6/Axl signaling plays an important role in phagocytosis, suppression of the inflammatory cytokine response, and final resolution of anti-inflammatory processes[4,59–61], thus it is not surprising that ablation of Axl signaling resulted in the overproduction of pro-inflammatory cytokines/chemokines in our disease setting.

Recent studies indicate that apoptotic cells can be involved in autoimmune processes[53,62]. Thus, it is conceivable that endothelial apoptosis and autoimmune injury may also occur secondary to R428-dependent BMPR2 decline. Interestingly, TAM family receptors heterodimerize with each other and with other RTKs, which often serve as an essential mode of their activation. An interaction of BMPR2 with VEGFR3 has been recently described in ECs[63]. Impairment of BMP signaling and exacerbation of hypoxia-induced PH has been determined in endothelial-specific VEGFR3 knock-out mice, proposing a novel mechanism of BMPR2 modulation in the context of PAH[63]. To the best of our knowledge, this is the first study demonstrating Axl as a key regulator of endothelial BMPR2 signaling and potential determinant of PAH emergence in humans. Our results demonstrate that although Axl can function as an oncogene in various cancers, it plays protective role against PAH development: (1) by halting the disease progression upholding the intrinsic mechanisms of EC survival; (2) by maintaining the BMP signaling in PAECs; and (3) by controlling the inflammation. Thus, Axl-mediated aggravation of PAH might be partly explained by proliferative/proinflammatory reprogramming of vascular wall

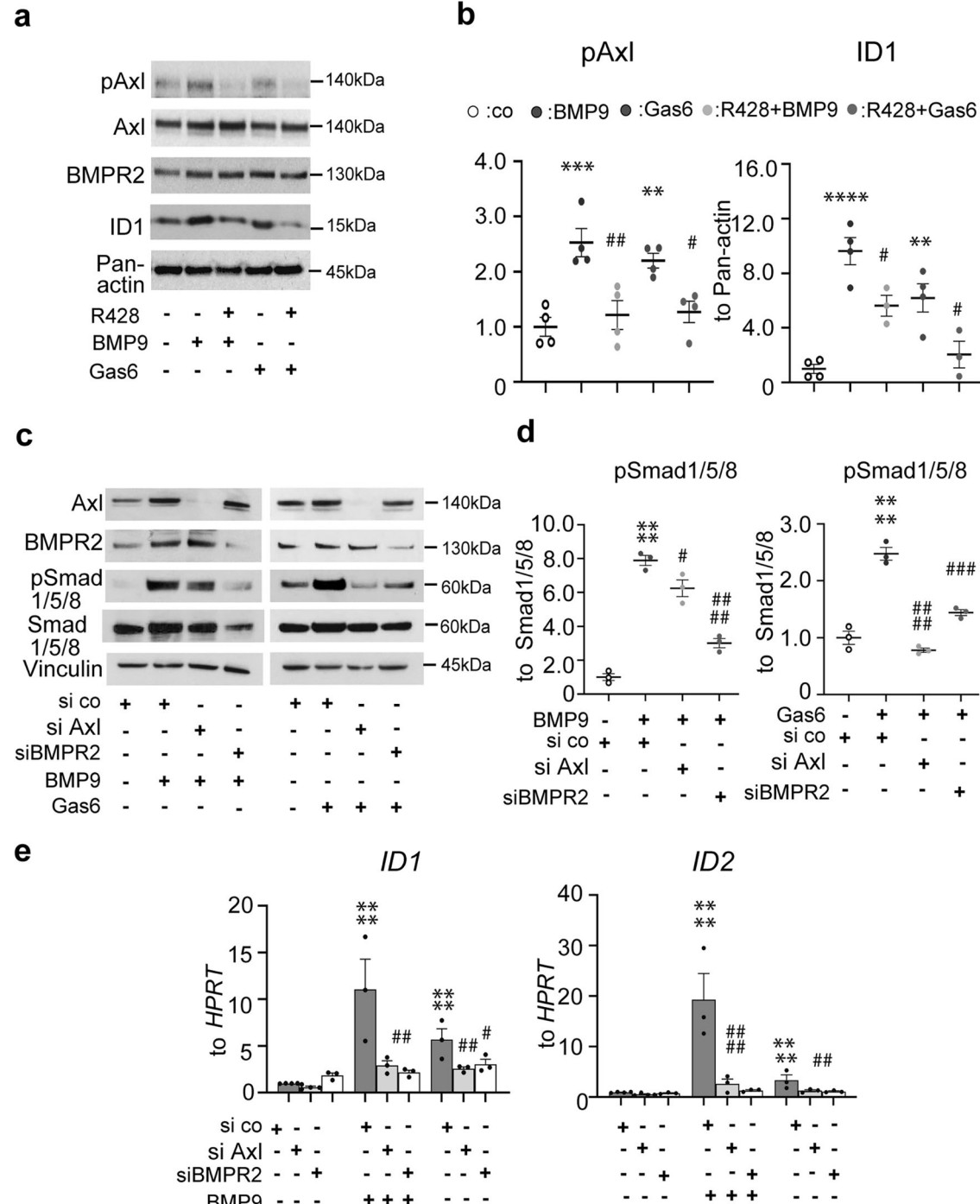

**Fig. 8 Modulation of bone morphogenetic receptor 2 (BMPR2) signaling pathway by Axl in human pulmonary arterial endothelial cells (hPAECs). a, b** Western blots and subsequent densitometry quantification of phospho-Axl, BMPR2, and ID1 after R428 treatment followed by Gas6 or BMP9 (2 h). Data from $n = 3$ biological independent experiments are presented as the $n$-fold change compared with DMSO-treated control (co) cells, defined with opened circles. Green and purple circles define Gas6 and BMP9. Light gray and dark gray circles define R428 + BMP9- and R428 + Gas6-treated conditions, respectively. **c, d** Representative western blots of Axl, BMPR2, and pSMAD1/5/8 and quantitative analyses of pSMAD1/5/8 expression upon the small interfering RNA (siRNA) treatment of Axl and BMPR2 in hPAECs. Opened circles define si scrambled (si co) cells, green and purple circles define BMP9 and Gas6 cells, respectively. Light gray and dark gray circles define siAxl and siBMPR2 cells, respectively. Data from $n = 3$ biological independent experiments are presented as the $n$-fold change compared with untreated control (co) cells. **e** Quantitative real-time PCR analyses of *ID1* and *ID2* after siRNA-mediated knock-down of Axl and BMPR2 and followed by BMP9 and Gas6 stimulations. Data from $n = 3$ biological independent experiments are presented as the $n$-fold change ($2^{-\Delta\Delta Ct}$) compared with scrambled siRNA control (si co). Statistical analyses were performed using one-way ANOVA with Tukey's or Newman-Keuls post hoc test for multiple comparisons. **a–e** ***$P < 0.001$, ****$P < 0.0001$ versus untreated scrambled siRNA controls (si co) or DMSO-treated control (co) cells; **a, b** #$P < 0.05$, ##$P < 0.01$ R428 + BMP9 and R428 + Gas6 versus R428; **c–e** #$P < 0.05$, ##$P < 0.01$, ###$P < 0.001$, ####$P < 0.0001$ for si co + BMP9, si co + Gas6 versus si Axl + Gas6/BMP9, and siBMPR2 + BMP9/Gas6. In a Pan-actin, in **c** Vinculin served as a loading control. All the data represent the mean ± SEM.

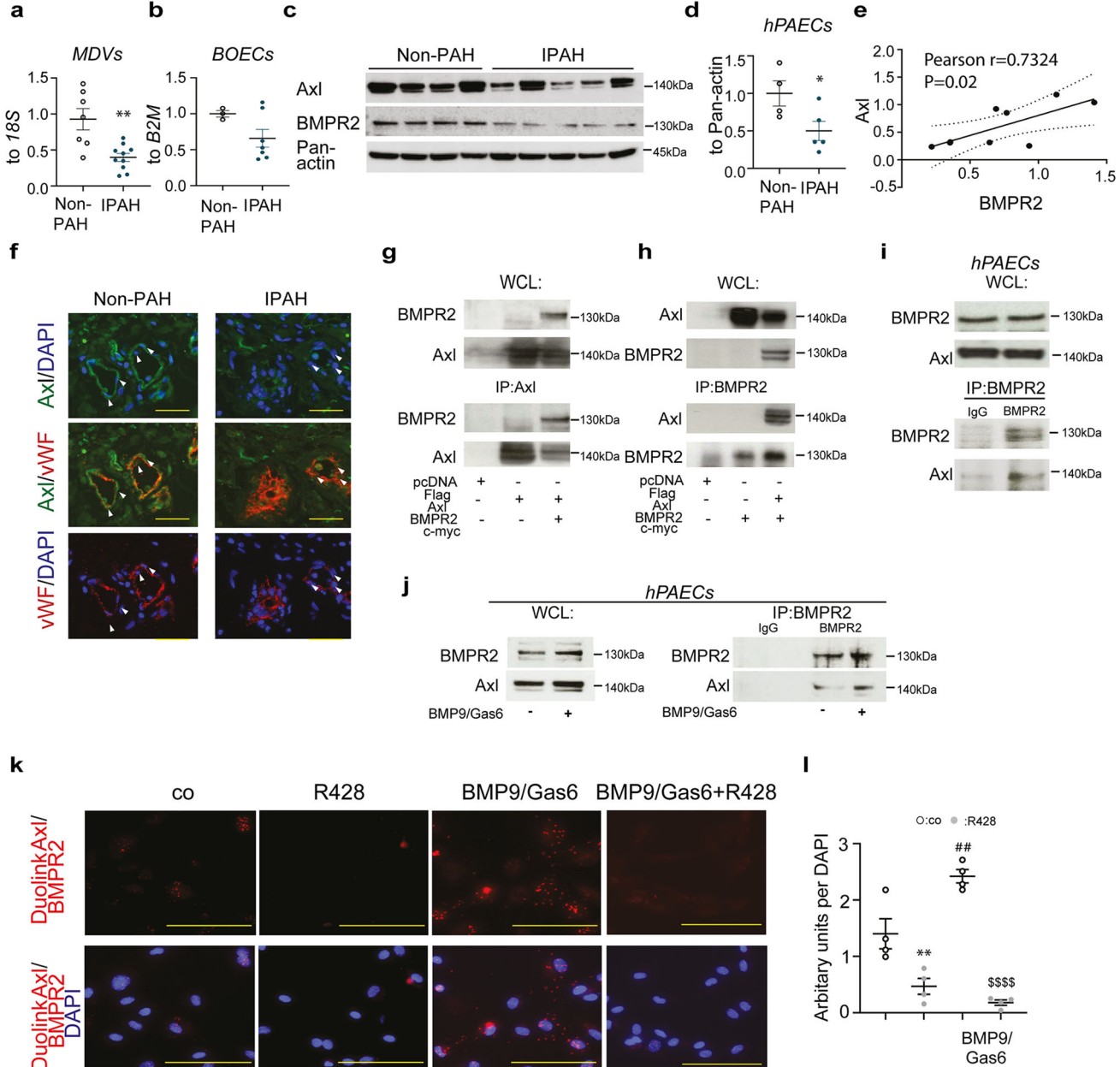

**Fig. 9 Axl closely associates with bone morphogenetic protein receptor 2 (BMPR2). a** Real-time quantitative PCR analyses of *AXL* mRNA expression normalized to *18S* in laser-assisted microdissected vessels (MDVs) from non-PAH biological independent controls (*n* = 7) and patients with IPAH (*n* = 10). **b** Real-time quantitative PCR analyses of *AXL* expression in mRNA from BOECs of non-PAH biological independent controls (*n* = 3) and patients with IPAH (*n* = 7), normalized to β2-microglobulin (*B2M*). **c** Western blot analysis and **d** subsequent densitometric quantification of Axl expression in human pulmonary arterial endothelial cells (hPAECs), isolated from biological independent patients with IPAH (*n* = 5) and non-PAH controls (*n* = 4). Pan-actin served as a loading control. **e** Analysis of the correlation between BMPR2 and Axl expression in hPAECs from patients with IPAH (*n* = 5) and non-PAH (*n* = 4) controls. Pearson correlation coefficient (*r* value) and the significance (*P* value) of correlation analysis is represented on the graph. **P < 0.01, *P < 0.5 versus non-PAH control (co). For statistical analyses Student's *t*-test and correlation analyses were applied. **f** Representative examples of immunofluorescence staining of Axl (green) expression in tissue sections of non-PAH (*n* = 4) and IPAH (*n* = 4) human lung specimens. Sections were double stained with von Willebrand factor (vWF) (red) antibody. White arrowheads depict the expression of Axl in PAECs. Nuclei were visualized using 4′,6′-diamidino-2-phenylindole (DAPI; blue). Scale bars: 50 μm. **g, h** In HEK293 cells overexpressing Flag-Axl and BMPR2-Myc, **g** co-immunoprecipitation using Flag antibody to target Axl protein pulled down BMPR2 protein, and **h** co-immunoprecipitation using BMPR2 antibody pulled down Axl protein. Representative pictures of three independent experiments are shown. **i** Co-immunoprecipitation using BMPR2 antibody pulled down endogenous Axl in hPAECs **j** co-immunoprecipitations after 48 h of hPAEC stimulation with BMP9/Gas6. WCL = whole-cell lysate; IP = immunoprecipitant. **k** Representative immunofluorescent images and **l** quantification of interaction between BMPR2 and Axl by Proximity Ligation (PLA) assay in hPAECs treated with R428 followed by BMP9/Gas6 stimulation. PLA shows close proximity of Axl and BMPR2 in hPAECs. Duolink Axl/BMPR2 = PLA fluorescence signal; Duolink Axl/BMPR2/DAPI = merged PLA/diamidino-2-phenylindole fluorescence signals. Scale bars: 50 μm. Images are representative of four independent experiments. Statistical analyses were performed using one-way ANOVA with Newman-Keuls post hoc test for multiple comparisons. **P < 0.01 and ##P < 0.001 versus DMSO-treated control (co) cells; $$$$P < 0.0001 versus BMP9/Gas6 stimulation. All the data represent the mean ± SEM.

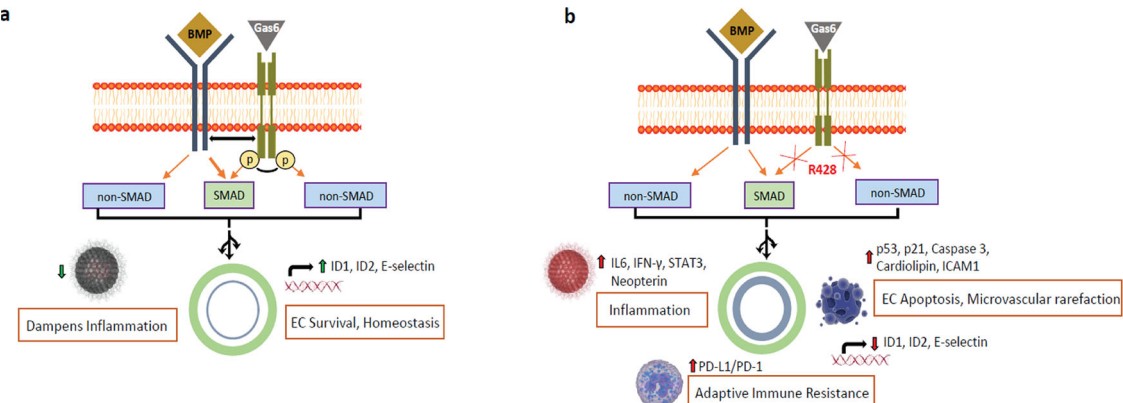

**Fig. 10 Proposed model of the effect of R428 on bone morphogenetic protein receptor (BMPR) 2 signaling and multiple facets of the inhibition of Gas6/Axl signaling in PAH. a** BMP signaling and EC homeostasis are maintained by Axl signaling. Gas6/Axl guards the anti-inflammatory response, EC survival and homeostasis by augmentation of BMPR2-related canonical SMAD. **b** The inhibition of Axl by R428 leads to the inhibition of canonical BMP/ SMAD signaling, augmentation of inflammation, adaptive immune evasion, EC apoptosis and microvascular rarefaction. ID1 = inhibitor of DNA binding 1; SMAD = mothers against decapentaplegic homolog 7; EC = endothelial cell; Gas6 = growth arrest-specific 6; IL6 = Interleukin 6; IFN-γ- Interferon gamma; ICAM1 = Intercellular Adhesion Molecule 1; PD-1/PD-L1 = programmed death ligand-1/programmed cell death-1.

cells, driven by BMPR2-related EC apoptosis (Fig. 10). Together, our results indicate that the systemic targeting of Axl in the context of PAH involves a number of potential pitfalls and may increase the risk of PH.

## Methods

**Human biomaterial.** Primary human PASMCs from healthy individuals were obtained either from Lonza (CC-2581 Basel, Switzerland) or from the UGMLC Giessen Biobank of the Justus-Liebig University Giessen (Giessen, Germany). HPASMCs from patients with PAH were obtained exclusively from the UGMLC Giessen Biobank. The human biomaterial collection was approved by the Ethics Committee of the Justus-Liebig University (ethics vote number 58/15). The patients have been informed and given their written consent for the use of biomaterials for research purposes. Human PASMCs were maintained in Smooth Muscle Cell Growth Medium-2 (SMC GM-2) (Lonza, CC-3182, Basel, Switzerland), provided with supplement mix, containing 5% fetal bovine serum (FBS), human basic fibroblast growth factor (hFGF-B, 2 ng/ml), epidermal growth factor (hEGF, 0.5 ng/ ml), and insulin (5 μg/ml). hPAECs from healthy individuals were purchased from PromoCell (C-12241, Heidelberg, Germany). The human lung specimens used for PAH-hPAEC cultures were obtained from patients with PAH during lung trans-plantation. PAH was diagnosed by cardiac catheterization at the National Refer-ence Center for PAH, in a program approved by our institutional ethics committee, Comité de Protection des Personnes Ile-de-France VII, and written informed consent for participation in the study was obtained (protocol N8CO-08- 003, ID RCB:2008-A00485-50). Lung histology was assessed by a pathologist, after trans-plantation, and confirmed the diagnosis of PAH. PH was defined as resting mean pulmonary arterial pressure ≥25 mmHg. Human PAECs were maintained in cul-ture with Endothelial Cell Growth Medium 2 (EGM-2), consisting of 2% FBS, 5 ng/ ml hEGF, 10 ng/ml hFGF-B, 20 ng/ml R3 IGF-1, 0.5 ng/ml VEGF, 1 μg/ml ascorbic acid, 22.5 μg/ml heparin, and 0.2 μg/ml hydrocortisone (PromoCell, C-22111, Heidelberg, Germany). Cells were maintained in complete EC growth medium-2 (EGM-2) and were used at passages 4–6. Blood outgrowth ECs (BOECs) were obtained and cultured as described previously[64,65]. All studies and procedures to obtain human specimens were directed according to the Declaration of Helsinki.

Both platelet-derived growth factor (PDGF-BB) and growth medium (GM), a mixture of growth factors were applied, as PAH stimuli for proliferation assays. For stimulation experiments, hPASMCs were starved in serum-free SMC Basal Medium (without supplement mix) for 24 h before each experiment and then treated with R428, as indicated. For the stimulation experiments hPAECs were starved for 20 h before each experiment in the medium M200 (M200500, Thermo Fisher Scientific, Waltham, USA) with addition of 2% FBS (AC-SM-0033, Anprotec, Bruckberg, Germany). Cells were exposed to R428 /BGB324 (HY-15150, MedChem Express, Monmouth Junction, USA) at the concentrations indicated in the figures. After one hour of R428 treatment, human PASMCs and PAECs were stimulated with recombinant PDGF-BB (30 ng/ml), SMC GM-2, Gas6 (200 ng/ml), or various BMPs (all at 10 ng/ml) either for 24 h, 48 h, or as indicated in the figures. For the determination of tyrosine phosphorylation of Axl by PathScan Phospho-Axl (panTyr) Sandwich ELISA assay, hPAECs were grown for 24 h, starved for 20 h and then stimulated with bone morphogenetic proteins (BMPs; all at 10 ng/ml) and growth arrest-specific protein 6 (Gas6; 200 mg/ml) for 2 h. All experiments were performed with cells between passages 5 and 7. Visual analysis was carried out by bright-field microscopy (Eclipse TS100; Nikon GmbH, Duesseldorf, Germany).

**Proliferation and migration assays.** Proliferation of human PASMCs was determined by measuring the incorporation of 5-bromo-2′-deoxyuridine (BrdU) with a colorimetric cell proliferation enzyme-linked immunosorbent assay (Cell Proliferation ELISA, BrdU kit) and by immunofluorescent analyses of the expression of proliferation marker Ki67 (ACK02, Leica, Wetzlar, Germany). Quantification of DNA synthesis by a colorimetric BrdU ELISA was carried out by absorbance measurement at 370 nm using 492 nm as a reference wavelength, and final graphs represent data from four individual experiments performed in tripli-cates. For analyses of Ki67 staining, 150–200 human PASMCs in five random fields of two different subpopulations were counted in each experiment. Proximity Ligation Assays (PLA) have been performed according to manufacturer's instructions (Duolink® Proximity Ligation kit, Merck, Darmstadt, Germany).

Two types of the migration experiments have been performed: the wound-healing assays and the transwell assays. The predefined wound areas for the wound-healing assays were created by removable cell culture inserts (Ibidi, Planegg/Martinsried, Germany). The wound areas were photographed before and after 16 h of incubation with PDGF-BB (30 ng/ml) with or without R428. The images were taken using a Leica inverted microscope (Leica, Wetzlar, Germany). The absence of proliferation was ensured by the addition of cytosine arabinoside (AraC). At 16 h after the completion of experiments, hPASMCs were fixed and stained with DAPI (4′,6′-diamidino-2-phenylindole, 0.5 μg/ml, Sigma, Munich, Germany). Migration rate was measured by quantifying the number of DAPI-positive hPASMCs that were migrated into the wound region. Results are given at an *n*-fold change normalized to the absorbance for DMSO-treated control cells after 24 h. The Infinite M200 Pro instrument from Tecan Group (Männedorf, Switzerland) was used to determine the absorbance at the desired wavelengths.

**Cloning of LeGO vectors and production of lentiviral particles.** Cloning of LeGO vector for overexpression of human Axl was performed based on the Len-tiviral gene ontology overexpression system (for detailed protocols and vector maps refer to http://www.lentigo-vectors.de). Full-length cDNA clone for Axl (pDONR223, #23945) was purchased from Addgene (Cambridge MA, USA). For overexpression, full-length cDNA from this vector was cloned into the LeGO-iG2-Puro⁺ vector using EcoRI and NotI restriction enzymes (New England Biolabs) according to standard protocols. Primers and probes were purchased from Applied Biosystems (https://products.appliedbiosystems.com). Sequences of primers used for cloning are as follows, forward primer: 5′-atatgaattccgccaccatggcgtggcggtgcc-3′, and reverse primer: 5′-atatgcggccgcctaggcaccatcctcctgcc-3′. For silencing Axl pLKO.1 vector containing shRNA Axl sequence 5′-ccggcgaaatcctctatgtcaa-catctcgagatgttgacatagaggatttcgttttt-3′ was purchased from Sigma MISSION. The complete sequence of shRNA Axl, including the U6 promoter, was cloned in a LeGO-Cer-BSD vector. PspOMI and XhoI (New England Biolabs) enzymes were used for these steps. Sequences of primers used for PCR amplification to enrich sequence from pLKO.1 vector are as follows, forward primer: 5′-acggtatcgatcac-gagactagccctcgagc-3′ and reverse primer: 5′-tactgccatttgtgtcgacgtcgagaattc-3′. PCR-amplified fragments of new vectors were verified by DNA sequencing. Over-expression Axl was confirmed by qPCR. Primers used for PCR amplification are described by the following assay IDs, human GAPDH (Hs99999905_m1) and human Axl (Hs01064444_m1). Only cells with 4-fold overexpression compared to control cells were used for experiments. After transduction of PASMC cells with fluorochrome-marked lentiviruses mediating Axl overexpression, cells were exposed to R428 treatments and cultured in standard conditions

**Transfection with Axl siRNA**. Human PAECs were transfected with siRNA against Axl using the SMARTpool: ON-TARGETplus AXL siRNA (10 nmol) (L-003104-00-0010), using the ON-TARGETplus Non-targeting Control Pool (D-001810-10-20). Shortly, the transfection was performed using the Dharma-FECT1™ (DharmaFECT1, GE Dharmacon). The siRNA/ DharmaFECT1™ complexes were diluted in the ratio of 9 µl per 6 cm dish in Opti-MEM I medium (Fa. Gibco, 31985-047). After 4 h of incubation, the medium was replaced by EGM-2 and the cells were kept for additional 20 h. The starvation prior to the stimulation was performed in M200 medium for 20 h, and then the cells were treated with BMP9 for the times described in the figure legends. The knock-down of the relevant RNA transcripts and protein were quantified using qPCR and western blotting analyses, respectively.

**Plasmids, transfection of HEK293 cells, and immunoprecipitations**. HEK293 cells were split into 10 cm dishes and grown for 24 h. Next day, cells were transfected AXL-FLAG (105933, Addgene, Watertown, USA) and of BMPR2-Myc plasmids using Turbofect transfection reagent (R0531, Thermo Scientific, Massachusetts, USA) as per manufacturer's instructions. Cells were washed with ice-cold DPBS without Ca and Mg (P04-36500, PAN Biotech, Aidenbach, Germany) 24 h post-transfection. Lysates were collected using IP lysis buffer (87787, Thermo Fisher Scientific, Waltham, USA) with the addition of Phenylmethylsulfonyl Fluoride (PMSF), Sodium Fluoride (NaF), Sodium Orthovanadate (Na3VO4), and cOmplete™ EDTA-free Protease Inhibitor Cocktail PIC (11836170001 Roche, Basel, Switzerland). Preclearing was performed with addition of 30 µl of protein A/G plus Agarose beads and IgG antibody for 1 h rotation at 4 °C. Then, 10% of protein lysate was separated for whole-cell lysates (WCL) and 700 µg of the total protein was used for immunoprecipitations. Lysates were incubated either with FLAG antibody, targeting Axl (1 µl per 50 µl of total protein) (F1804, Sigma-Aldrich Chemie GmbH, Munich, Germany) or with BMPR2 antibody (10 µg per 1 mg of total protein) (sc-393304, Santa Cruz Biotechnology, Dallas, USA) for 1 h rotation at 4 °C. After 1 h of incubation, 40 µl of protein A/G-agarose beads was added to the antibody–lysate mixture and left overnight for rotation at 4 °C. Next day, the beads were washed with ice-cold IP lysis buffer and twice with PBS. Beads were boiled in 50 ml of 4X LDS sample buffer and 10X NuPAGE Sample Reducing Agent and the solubilized material underwent gel electrophoresis and western blot analysis. For immunoprecipitation of hPAECs, cells were split into 10 cm dishes and grown for 48 h in EGM-2 medium. After overnight incubation, cells were washed with ice-cold DPBS without Ca and Mg (P04-36500, PAN Biotech, Aidenbach, Germany) and lysed in IP lysis buffer. Lysis was performed by passing the lysates through the needle (20 G). The lysates were incubated with BMPR2 IP antibody (10 µg per 1 mg of protein), as described above. As a negative control, the pull down with IgG1 antibody (10 µg per 1 mg of protein) (sc-2025, Santa Cruz Biotechnology, Dallas, USA) was performed.

**Immunohistostaining and immunofluorescence**. Explanted human and rat lungs were flushed with saline, fixed in 4% PFA, and washed in PBS before dehydrations. Then the lungs were embedded in paraffin and 2–3 µm thick sections were obtained, hematoxylin/eosin staining and immunohistochemistry were performed with a microtome. From the rat studies, the left lobe of the lung was utilized for the paraffin embedding. Axl receptor has been detected on rat sections by the immunohistochemistry with the antibody from Antibodies Online (ABIN756022, Aachen, Germany) and on rat sections using the antibody purchased from Cell Signaling (C89E7); macrophages have been stained with monoclonal anti-CD68 antibody clone ED1 (MCA341GA, Bio-Rad Laboratories GmbH, Feldkirchen, Germany). For antigen retrieval after dehydration, rat lung blocks were cooked in rodent decloaker (Biocare medical by Zytomed Systems GmbH). For blocking of endogenous rat IgG and non-specific background in rat tissues, Rodent block R has been utilized (Biocare medical by Zytomed Systems GmbH). For antigen retrieval of human sections, Tris-EDTA (pH 9.0) or Citrate buffer (pH 6.0) were applied. After overnight incubation with all the antibodies, stained sections were extensively washed in either PBS or TBS, and antibody binding was determined using the ZytoChem Plus phosphatase polymer kit (Zytomed Systems GmbH, Berlin, Germany). The Warp Red Chromogen substrate kit (Zytomed Systems GmbH) was used where positive stain was in purple. The assessment of the medial wall thickness and the ratio of neointima/media was achieved by Elastica-van-Gieson staining. All the arteries were categorized according to their external diameter using a computerized morphometric analysis system (QWin: Leica, Wetzlar, Germany). Small arteries included those with an external diameter between 20 and 50 µm that have been used for the quantitative analyses. All the pulmonary vessels per each lung with a diameter of 20–50 µm were analyzed with a microscope at a 63-fold magnification. The medial wall thickness was calculated by using the following formula $MWT = (2 \times WT/EVD) \times 100$, where WT is wall thickness (the mean distance between the lamina elastic externa and the vessel lumen) and EVD is external vessel diameter. For the analyses of neointima/media ratio, neointima, the cellular layer between the lamina elastic interna and the pulmonary vessel lumen was divided by the media, the area between the lamina elastica interna and the lamina elastica externa. For double immunofluorescence staining of human lung cryopreserved sections, anti-goat Axl antibody (AF854, R&D Systems, Minneapolis, USA) was combined with anti-mouse α-SMA antibody (clone 1A4) (F3777, Sigma-Aldrich, St. Louis, USA) or anti-rabbit von Willebrand factor (vWF)

(ab9378, Abcam, Cambridge, UK). For immunofluorescence staining of PMVECs, the Isolectin GS-IB4 (from *Griffonia simplicifolia*) (I21411, Thermo Fisher Scientific, Waltham, USA) was applied.

**Permeability assay**. Human PAECs were split at 25,000 cells per insert. Cells were grown for several days in EGM-2 at 37 °C and 5% of $CO_2$. The medium was changed every second day. After cell confluency was reached, hPAECs were starved for 24 h in Endothelial Cell Basal Medium with the addition of 0.2% FCS. Following serum-starvation, cells were treated with R428 (1 µM), or TNF-α (5 ng/ml), or DMSO for the subsequent 24 h. Inserts were then transferred to a new receiver tray and a working solution of FITC-Dextran (1:40 diluted in the medium) (ECM644; Millipore, Bedford, MA) was added to the top chamber. Thereafter, light protected incubation was continued. Inserts were temporarily removed to another tray and 100 µl was taken to a special black 96-well opaque plate for fluorescence measurement. To determine the absorbance at the desired wavelengths (an excitation 485 nm and emission 535 nm), samples were measured every 6 h using an Infinite M200 Pro instrument from Tecan Group (Männedorf, Switzerland). Experiments were performed in quadruplicate.

**Western blot**. Total protein from rat lung tissues was extracted using cell lysis buffer (Cell Signaling Technology, MA, USA) using the TissueLyser LT (Qiagen, Germany) for homogenization of the samples. Protein lysates from human PASMCs and PAECs were isolated using RIPA buffer (Thermo Fisher Scientific). Both buffers contained protease and phosphatase inhibitors (Calbiochem, Carlsbad, Germany). After centrifugation at 12,000g for 10 min at 4 °C, protein concentration was estimated and normalized using a Bio-Rad protein assay kit (Bio-Rad Laboratories, Hercules, CA, USA). After denaturation for 10 min at 95 °C, equal amounts of protein (50 µg) were boiled in the presence of NuPAGE LDS Sample Buffer and NuPAGE Reducing Reagent and loaded on to 4–12% NuPage Bis-Tris Gels (Invitrogen, Carlsbad, CA, USA). Gel electrophoresis was followed by transfer to nitrocellulose membrane. Blocking was performed in 5% non-fat dry milk dissolved in 1% TBS/Tween-20 (TBS/T) at RT. Primary antibodies were incubated for overnight at 4 °C in 5% BSA in 1% TBS/T. All primary antibodies were diluted in 5% BSA diluted in Tris-buffered saline/Tween-20 for 1 h at room temperature and then incubated with primary antibodies at 4 °C overnight with gentle agitation. All primary antibodies used for western blot analyses are represented in Supplementary Table 1.

*Amersham HRP-labeled secondary antibodies*. The ECL Mouse IgG, HRP-linked whole Ab from sheep (NA931) and the ECL Rabbit IgG, HRP-linked whole Ab from donkey (NA934) purchased from GE Healthcare (Little Chalfont, Buckinghamshire, England) were diluted 1:10,000 in 5% non-fat dry milk (T145.3, Carl Roth, Karlsruhe, Germany) in 1% TBS/T. The Anti Goat IgG HRP antibody was obtained from Sigma-Aldrich (A8919, Sigma-Aldrich, St. Louis, USA). After 1 h of incubation at RT and washing in 1% TBS/T, the membranes were incubated in ECL Prime Western Blot Detection Solution (GE Healthcare, Little Chalfont, and Buckinghamshire, England) for visualization of the protein bands.

**RNA isolation, cDNA synthesis, quantitative RT-PCR**. Total RNA from rat lung tissues was isolated using TRIzol reagent (Thermo Fisher Scientific, Waltham, USA) according to the manufacturer's protocol. RNA from PASMCs and PAECs was isolated using the RNeasy Kit (Qiagen, Hilden, Germany) including an on-column DNase digest step according to the manufacturer's protocol. RNA concentration was measured via Nanodrop ND1000 (Thermo Fisher Scientific). cDNA was generated by standard reverse transcription that was performed including 1 µg RNA, dNTPs, Random Hexamer Primer (Thermo Fisher Scientific, Waltham, US SO142), and M-MLV reverse transcriptase (Ambion, Austin, US, AM2044) in a thermocycler. The PCR products were detected on 96-well plates, PCR was performed on an Mx3000P® qPCR System machine (Stratagene, Agilent Technologies, Waldbronn, Germany) using iTaq™ Universal SYBR® Green Supermix (Bio-Rad Laboratories, Hercules, CA, USA), primers (Metabion International AG, Planegg/Steinkirchen, München, Germany), RNAse-free water (Qiagen, Hilden, Germany), and 1 µl of cDNA samples. Ct values were normalized to the reference genes *GAPDH, 18S, HPRT*, and *B2M* and relative gene expression was calculated on the basis of ΔCt values. The ΔCt values for each target gene were calculated by $\Delta Ct = Ct_{housekeeping\ gene} − Ct_{target\ gene}$. Each gene was normalized to a housekeeping control as indicated. Primer sequences used for quantitative real-time PCR analyses are represented in Supplementary Table 2.

**ELISAs and TUNEL**. All the enzyme-linked immunosorbent assays for the study were performed according to manufacturer's instructions. The In Situ Cell Death Detection Kit TMR red has been utilized for the paraffin-embedded (3 µm) thick rat lung sections following the recommendations by the company (12156792910, Roche, Basel, Switzerland). Recombinant proteins and ELISA kits used for the study are represented in Supplementary Table 3.

**Animal experiments**. All animal experiments were approved by the local authorities and the federal authorities for animal research of the Regierungspräsidium Giessen (approval numbers: GI 20/10 Nr. G 50/2016 and GI 20/10 Nr.33/3013).

Adult male Wistar-Kyoto rats (220–250 g body weight) and adult male Sprague-Dawley rats (300–350 g body weight) were obtained from Charles River GmbH (Sulzfeld, Germany). For the MCT model of PH, Sprague-Dawley rats were subcutaneously injected with MCT (60 mg/kg body weight) and randomized on day 21 to receive a daily dose of R428 (100 mg/kg body weight) or vehicle (1% of methylcellulose) by oral gavage during the subsequent 2 weeks. Rats injected with saline instead of MCT on day 0 were used as healthy controls. For the Sugen 5416 hypoxia (SuHOX) rat model of PH, Wistar-Kyoto rats were injected with Sugen 5416 (20 mg/kg body weight) and kept in hypoxic chambers (10% oxygen). After 3 weeks of hypoxic exposure, rats were re-exposed to normoxic conditions for 2 additional weeks and randomized to receive a daily dose of R428 (100 mg/kg body weight) or vehicle by oral gavage. Rats injected with vehicle (100% dimethyl sulfoxide) were used as hypoxic controls. Transthoracic echocardiography was performed with the Vevo 2100 system (VisualSonics, Toronto, Canada) equipped with an 18–38 MHz transducer (MS400, for mouse cardiovascular assessment). Measurement of RVSP and systemic arterial pressure was performed as described previously[66,67].

**Statistics and reproducibility**. Statistical analysis was performed using Prism 8.00 (GraphPad, San Diego, CA, USA). Biological replicates are defined as different animals or cultured cells under the same experimental conditions but processed in parallel. Independent cell culture experiments are defined as different sets of cells processed at different times. No statistics were derived from technical replicates. All data are expressed as mean ± SEM. For comparison of two groups, Student's $t$-test was applied; for comparisons involving more than two groups, one-way analyses of variance corrected by Tukey's or Newman-Keuls post hoc tests were applied. $P < 0.05$ was considered statistically significant.

**Reporting summary**. Further information on research design is available in the Nature Research Reporting Summary linked to this article.

## Data availability

All data generated in this study are included in the article (and its Supplementary Data files). Source data for figures and supplementary figures can be found in Supplementary Data 1. Original blots/gel images are shown in Supplementary Data 2. The data underlying all findings of this study are available from the corresponding author upon reasonable request and are provided as a separate source data file.

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

## Acknowledgements

Funded by the Deutsche Forschungsgemeinschaft (DFG, German Research Foundation) – Projektnummer 268555672 - SFB1213, projects A08, B04, and CP02. P.D.U. was supported through a BHF Programme grant to N.W.M. (RG/13/4/30107). N.W.M. is a BHF Professor and NIHR Senior Investigator. The authors thank Ewa Bienek, Christina Vroom, Sophia Hattersohl, Stephanie Viehmann, and David Dippel (Excellence Cluster Cardio-Pulmonary System, Universities of Giessen and Marburg Lung Center, Member of the German Center for Lung Research (DZL), Justus-Liebig-University, Giessen, Germany) for their technical support. We also thank Bruno Poettker and Ingrid Henneke for assistance in writing of the animal experiment proposals. Claire Mulligan (Beacon Medical Communications Ltd, Brighton, UK) provided editorial support, funded by the University of Giessen.

## Author contributions

T.N. and R.T.S. designed experiments, analyzed and interpreted data, and wrote the manuscript. T.N., N.R., B.K., S.V., I.B-B., P.C., M.S., N.P., E.G., C.L., S.H. and G.M. performed the experiments, analyzed and interpreted data. F.P., H.G., H.A.G., N.W., J.W., M.W., P.D.U., S.L., N.W.M., W.S. and R.T.S. analyzed and interpreted data. All authors were involved in reading and critically revising the manuscript.

## Funding

## Competing interests

P.D.U. is a founder of, and scientific advisor to Morphogen-IX Ltd. N.W.M. is a founder and CEO of Morphogen-IX Ltd. P.D.U. and N.W.M. have published US (US10336800) and EU (EP3166628B1) patents entitled: "Therapeutic Use of Bone Morphogenetic Proteins". All other authors declare no competing interests.
