## [Transparent Peer Review File. · Communications Biology]

Reviewers' comments:

Reviewer #1 (Remarks to the Author):

In this study the authors investigate the role of the receptor tyrosine kinase Axl, both by loss- and gain-of-function studies. Axl is a receptor tyrosine kinase. Its expression declines in endothelium of patients with PAH.

Axl is the receptor of Gas6, and this signalling pathway has been associated with cancer metastasis, and favouring a tumor-promoting environment and poor prognosis. In this study Axl is studied in the context of Pulmonary Arterial Hypertension (PAH). The authors demonstrate a protective role of Gas6/Axl in PAH and highlight an interaction between the BMP and Gas6/Axl signalling pathways. Remarkably, Axl inhibition augments both apoptotic and inflammatory responses in experimental PAH, and inhibits BMPR2 signalling. Axl and BMPR2 interact endogenously, and in a BMP9/Gas6 dependent manner. This interaction protects against PAH and dampens inflammation in lung vasculature. Axl is an important new tuner of BMP signalling, and both BMP and Gas6 ligands can activate the other pathway. In this original MS, the authors document extensively a new BMPR2 co-receptor that is protective against PAH. The work seems robust?

Remarks

The abstract is difficult to read (chaotic), this contrasts to the introduction which is written very clearly. In the abstract, it is often unclear what are results from in vivo studies, and what are cell culture data. Consider a restructuring of the abstract.

ICAM1 mRNA is increased upon R428 treatment or Axl KD. Is VCAM mRNA similarly affected?

Special features of both disease models (Monocreatin and Sugen5416/hypoxia) need to be discussed in more detail.

Reduced BMPR2 levels, both mRNA and protein, can be expected to reduce downstream signalling. Is the Axl-BMPR2 interaction affecting primarily BMPR2 protein stability and location at the cell surface, and does this result in higher activation levels of downstream components. Or is decreased BMPR2 expression causing the reduced levels of BMPR2 protein and signalling. How do you explain the reduced BMPR2 mRNA levels?

Suppl fig 3: figure not well visible

Fig. 6 is very overloaded, consider moving some parts to supplementary figures. Isolectin staining is not well visible.

Fig. 8 labelling of panels "I" and "j" is missing

Typo's

Ln 218 "revel" should read "reveal"

Reviewer #2 (Remarks to the Author):

This manuscript by Novoyatleva and co-workers explores a potential role for Axl in pulmonary arterial hypertension. Axl is reported to be marker for EMT and involved in tumor cell growth and survival, all features reported to be involved in the pathogenesis of PAH. A possible link between Gas6/Axl has been suggested before, but this is the first in vivo study.

The authors show in an elegant series of experiments combining human and rat cells that Axl inhibition using R428 or knockdown using siRNAs reduces SMC and endothelial cell proliferation and migration in vitro. Using two animal models for experimentally induced PAH, the authors show that R428 treatment worsened disease progression in these rats. Finally, to unravel the mechanism of action, the authors show that Axl inhibition interfered with BMP9 induced pSmad1/5/8 in PAECs. Interestingly, Axl was found to be associated with the BMPR2 to facilitate ligand binding.

This is an interesting observation and possible new target for treatment, but there are some issues

that need to be addressed.

- Previous studies have reported that 1) hypoxia stabilizes the Gas6/Axl signaling and 2) overexpression of Axl reduces EC function in a hypoxic glycemc environment. In this study, the authors use two experimental PAH models, MCT and Sugen hypoxia. When analyzing the data in suppl figure 1, the induction of Axl is more pronounced in SuHOX when compared to MCT. This might be related to hypoxia/hif-1 axis. What is the impact of hypoxia on the mechanism proposed? Taking this into consideration, the data presented in suppl 1e,f should also be done for SuHox?

- In an elegant series of in vivo experiments, the authors clearly show that R428 deteriorates disease progress in these two animal models. What is not clear for this study is the impact R428 has in control animals as there might be side effects at the dosis given, especially as two different rat strains are used.

- Gas6/Axl has been linked in literature to modulate inflammation and fibrosis, two processes contributing to PH progression. Does inhibiting Axl modulate fibrosis and inflammation in the animals?

- The authors show that treating PAECs in vitro with R428 increases ICAM-1 expression in PAECs. This might hamper barrier function and an explanation why the animals do worse. What is the impact of Axl modulation on barrier function?

- Axl has also been reported to modulate TGFbeta signaling. At 1 uM, R428 is reported to prevent TGFbeta induced invation. As aberrant TGFbeta signaling is part of the PAH pathology, how is this pathway influenced and contribute to the phenotype observed? Is pSmad1/5 and pSmad2/3 influenced in the R428 treated animals and does this correlate with reduced Axl in those cells.

- Figure 1 shows the effect of Axl inhibition on SMCs. In iPAH SMC the concentration needed of R428 is higher comparted to control cells. Is this related to the expression level of Axl and the levels of BMPR2 in the cells.

- In figure 2, the authors analyze the impact on EC behavior. These cells are extremely sensitive for R428. Is this related to Axl levels and BMPR2 expression. Are iPAH patients ECs as sensitive?

- In figure 3, the authors show the impact of R428 on experimentally induced PH. Firstly, the figure legend is not correct, a-e is MCT and f-j is sugen. Secondly, the impact of Axl inhibition is more clear in MCT compared to SuHOX. Can the authors explain this? Does this reflect hypoxia induced vs inflammation/liver induced disease progression? What would be most predictable for the patient?

- In figure 4, the impact of R428 in SuHOX lungs is analyzed in more detail. Which cell type is affected?

- The authors nicely show that R428 modulates BMPR2 levels in lung tissue of suHOX animals. Recent studies have shown that activin and activin receptors are involved in PAH. How does R428 influence this pathway?

- In the last part of the paper, the authors aim to unravel the mechanism of action, and suggest that Axl interacts with BMPR2. In figure 7/8 the authors show that Axl and R428 interfere with BMP/SMAD1/5 signaling by modulating BMPR2 levels. This is an interesting mechanism, but this reviewer has some concerns. The effect seen treating the cells with R428 could very well be indirect. R428 induces a proinflammatory phenotype, demonstrated by the high iCAM and high IL6. Previous work, also by some of the co-authors, have demonstrated that this eventually results in BMPR2 downregulation. Therefore more solid validation is needed to demonstrate that this is a direct effect.

1. The concentration of BMP9 and Gas6 used to stimulate the cells is very high. If Gas6 is contaminated with some kind of BMP, you can have the results shown. Therefore, a neutralizing antibody should be used in combination with Gas6.
2. Neither BMP9 nor Gas6 were able to induce BMPR2 expression in fig 7K and Id1 and pSmad1/5 should be done in the presence of siRNA for BMPR2 and ACVR2A/B.
3. Figure 7G lacks a BMP control.
4. In figure 8, to show interaction, the IP performed should be repeated with BMP9 and Gas6 to draw this conclusion.
5. The PLA signals in fig 8i is not clear. Not all signal seems to be at the cell membrane.

Minor comment

- As not all samples are shown by WB, it would be informative if dots were shown in the bar graph as was done in figure 3. Not all legends show the number of samples analyzed.
- Molecular weights need to be indicated for the western blots.
- Suppl figure 5 A: pStat3/ stat3 and pan-actin only have 13 lanes while Nitro and NFkB have 14 lanes.
- Suppl Figure 6C, pan actin has 14 lanes while pd1 only 13.
- Suppl figure 6H: from the images shown, there is more proliferation in SuHOX with R428.

Reviewers' comments:

Reviewer #1 (Remarks to the Author):

In this study the authors investigate the role of the receptor tyrosine kinase Axl, both by loss- and gain-of-function studies. Axl is a receptor tyrosine kinase. Its expression declines in endothelium of patients with PAH.

Axl is the receptor of Gas6, and this signalling pathway has been associated with cancer metastasis, and favouring a tumor-promoting environment and poor prognosis. In this study Axl is studied in the context of Pulmonary Arterial Hypertension (PAH). The authors demonstrate a protective role of Gas6/Axl in PAH and highlight and interaction between the BMP and Gas6/Axl signalling pathways. Remarkably, Axl inhibition augments both apoptotic and inflammatory responses in experimental PAH, and inhibits BMPR2 signalling. Axl and BMPR2 interact endogenously, and in a BMP9/Gas6 dependent manner. This interaction protects against PAH and dampens inflammation in lung vasculature. Axl is an important new tuner of BMP signalling, and both BMP and Gas6 ligands can activate the other pathway. In this original MS, the authors document extensively a new BMPR2 co-receptor that is protective against PAH. The work seems robust?

Remarks

The abstract is difficult to read (chaotic), this contrasts to the introduction which is written very clearly. In the abstract, it is often unclear what are results from in vivo studies, and what are cell culture data. Consider a restructuring of the abstract.

We are thankful for this comment. The abstract has been restructured.

ICAM1 mRNA is increased upon R428 treatment or Axl KD. Is VCAM mRNA similarly affected?

The expression of VCAM1, similarly to ICAM1, was enhanced by R428 treatment. Conversely, Axl KD promoted a decrease in VCAM1 mRNA, suggesting that the small molecular inhibitor R428 might contribute to some distinct molecular mechanisms, in altering the expression of these pro-inflammatory molecules. The new data has been incorporated into the text of the Results part. Please see new sentence on Page 5, Lines 151-157. "Thus next we aimed to investigate the effect of Axl blockage on markers of endothelial dysfunction and damage/vascular injury in hPAECs. R428 treatment and Axl knock-down each substantially increased mRNA levels of intercellular adhesion molecule 1 (ICAM1) (Figure 2^{f,h}). Interestingly, R428 treatment and Axl deletion provided opposing results on the expression of vascular cell adhesion molecule 1 (VCAM1), indicating that small molecular inhibitor of Axl signaling pathway and Axl deletion might contribute to non-overlapping molecular signaling pathways (Figure 2^{g-h}).

Special features of both disease models (Monocreatin and Sugen5416/hypoxia) need to be discussed in more detail.

The following sentences have been added to the discussion part. Please see Page 11, Lines 292-303 "Both MCT and SU5416/hypoxia animal models have been employed for decades to develop experimental PAH, by inducing PH and RVH. MCT, a highly toxic alkaloid, when ingested stimulates widespread pneumotoxicity, connected with increased cell proliferation and systemic inflammation (PMID: 21964406). In the SuHX model of pulmonary hypertension, SU5416 mediated VEGFR2 blockade, in combination with normobaric hypoxia, induces a severe PH with elements of inflammation and angio-obliteration (PMID: 25705569). Both models are characterized by EC injury that is associated with BMPR2 dysregulation and BMP signaling deficiency. Interestingly, in

contrast to SU5416/hypoxia model of severe PAH, MCT model does not exhibit endothelial cell-mediated angio-obliteration, but reveals pronounced pulmonary arterial hypertrophy (PMID: 21964406). In our study Axl inhibition in MCT-injected rats resulted in a significant increase of the neointima/media ratio, suggesting that EC dysfunction in this model is a direct consequence of EC damage.

Reduced BMPR2 levels, both mRNA and protein, can be expected to reduce downstream signalling. Is the Axl-BMPR2 interaction affecting primarily BMPR2 protein stability and location at the cell surface, and does this result in higher activation levels of downstream components. Or is decreased BMPR2 expression causing the reduced levels of BMPR2 protein and signalling. How do you explain the reduced BMPR2 mRNA levels?

We would like to mention that no changes in the expression of BMPR2 mRNA were noted upon R428 treatment in hPAECs 24 hours after the drug treatment, suggesting that in ECs the inhibition of Axl is rather directly altering BMPR2 protein stability resulting in deficiency of the BMPR2 signaling pathway. Interestingly, R428 treatment of hPASCs (but not hPAECs) resulted in a strong decline of both mRNA and protein expression, which can be explained transcriptional regulation of miRNAs. Various miRNAs (as miR21, miR17) can downregulate BMPR2 mRNA transcription under a diverse set of circumstances. Axl inhibition by R428 has been shown to enhance caspase-dependent TRAIL-induced apoptosis of cancer cells, through the increase of miR-708 (PMID: 31269715). Interestingly expressional analyses of different miRNAs in patients with PH, revealed significant upregulation of miR-708 (PMID:27214554). Thus we suspect that Axl-miRNA708-BMPR2 axis potentially may serve as an additional molecular mechanism, explaining BMPR2 mRNA decrease specifically in hPASCs.

Suppl fig 3: figure not well visible

We have magnified the images on Suppl fig 3.

Fig. 6 is very overloaded, consider moving some parts to supplementary figures.

We have followed the suggestions and transferred two graphs into the new Supplemental Figure 7.

Isolectin staining is not well visible.

The existing images on Figure 6 were modified.

Fig. 8 labelling of panels "I" and "j" is missing

The missing labels were added on Figure 8.

Typo's

Ln 218 "revel" should read "reveal"

The typo has been corrected.

Reviewer #2 (Remarks to the Author):

This manuscript by Novoyatleva and co-workers explores a potential role for Axl in pulmonary arterial hypertension. Axl is reported to be marker for EMT and involved in tumor cell growth and survival, all features reported to be involved in the pathogenesis of PAH. A possible link between Gas6/Axl has been suggested before, but this is the first in vivo study.

The authors show in an elegant series of experiments combining human and rat cells that Axl inhibition using R428 or knockdown using siRNAs reduces SMC and endothelial cell proliferation and migration in vitro. Using two animal models for experimentally induced PAH, the authors show that R428 treatment worsened disease progression in these rats. Finally, to unravel the mechanism of action, the authors show that Axl inhibition interfered with BMP9 induced pSmad1/5/8 in PAECs. Interestingly, Axl was found to associated with the BMPR2 to facilitated ligand binding.

This is an interesting observation and possible new target for treatment, but there are some issues that need to be addressed.

- Previous studies have reported that 1) hypoxia stabilizes the Gas6/Axl signaling and 2) overexpression of Axl reduces EC function in a hypoxic glycemc environment. In this study, the authors use two experimental PAH models, MCT and Sugen hypoxia. When analyzing the data in suppl figure 1, the induction of Axl is more pronounced in SuHOX when compared to MCT. This might be related to hypoxia/hif-1 axis. What is the impact of hypoxia on the mechanism proposed? Taking this into consideration, the data presented in suppl 1e,f should also be done for SuHox?

We are thankful for this comment. To clarify, whether hypoxia is indeed playing an essential role in Axl regulation in the context of PH, we determined the expression profile of Axl receptor in chronic-hypoxia induced mouse model of PH. Immunohistochemical analysis of mouse lung sections indicated an augmented receptor expression in the medial layer of pulmonary arteries of murine lungs (please, see new Supplementary Figure ^{1e} included). The mRNA analysis of Axl expression in Laser-assisted microdissected vessels of mouse lung sections exhibited an almost 7-fold of Axl increase (Please, see new Supplemental Figure ^{1g}). Furthermore, Axl protein expression profile of total lung homogenates of chronic-hypoxia exposed mice indicated an almost 5-fold of Axl overexpression (new Supplementary Figure ^{1f}). To confirm that hypoxia induces Axl expression in cultured mouse PSMCs, the mRNA analysis of Axl is implemented in the revised version of the manuscript. An augmented expression of the receptor was also confirmed in mouse PSMCs (Supplemental Figure ^{1h}). Our data suggest that Axl expression is driven by hypoxia both in chronic-hypoxic induced PH and hypoxia-exposed PSMCs. Please see the new data included in Supplemental Figure 1 (new panels: 1e, 1f and 1g [left] and 1h) and the new text incorporated. See Page 4, Line 121” The expression of Axl was augmented in the Lungs of SuHx and MCT rat models, and a chronic-hypoxia mouse model of PH, as compared to controls (Supplementary Fig. ¹) and Line 123 “As Axl expression was strongly induced in pulmonary vessels (Supplementary Fig. ^{1g}) and pulmonary arterial smooth muscle cells (PSMCs) in experimental PH (Supplementary Fig. ^{1h}), we explored the functional effect of Axl inhibition on hPASM C proliferation and migration using a clinically tested small molecule inhibitor R428 (BGB324)“. We also added the following lines into the text of the Discussion part. Please, see page 12, Lines 311-312: “In contrast, the expression of Axl was strongly induced in rat and mouse PH-PASMCs”.

- In an elegant series of in vivo experiments, the authors clearly show that R428 deteriorates disease progress in these two animal models. What is not clear for this study is the impact

R428 has in control animals as there might be side effects at the dosis given, especially as two different rat strains are used.

Due to the Covid19 induced Laboratory restrictions in our animal facility, the requested animal experiments addressing the effects of Axl inhibition in healthy animals could not be performed. However, we believe that these experiments will not significantly contribute to the main content of the manuscript.

- Gas6/Axl has been linked in literature to modulate inflammation and fibrosis, two processes contributing to PH progression. Does inhibiting Axl modulate fibrosis and inflammation in the animals?

*We already provided the descriptive analyses of various pro-inflammatory factors, as Nitrotyrosine, markers of STAT-pathway (phospho-STAT3, SOC3), cytokines (IFN- γ , IL-6) and chemokines (CXCL10, CXCL11) in the Lungs of both SuHOX and MCT models of PH. A marked and significant increase of numerous pro-inflammatory mediators of PAH has been detected in experimental rat PAH. Concomitantly, we also determined the accumulation in the number of CD68-positive macrophages in lung sections of both SuHOX- and MCT-treated animals. Overall, our data indicate that Axl modulation augmented a pro-inflammatory response in the PAH context. To determine, whether Axl inhibition also modulates a fibrotic response, we now performed collagen assay analyses of rat Lungs using the **Sircol™ Collagen Assay** kit, which allows measuring collagen production, indicative of in vitro extracellular matrix formation. Interestingly, R428 administration of both MCT and SuHx-treated rats did not alleviate collagen production, indicating that Axl inhibition does not majorly impact fibrosis in both experimental models of PAH. Please, see the graph below.*

Figure. Quantitative analysis of total collagen amount (Types 1 to V) in rat lungs isolated from **a** MCT and **b** SuHOX experimental PAH, monitored by Sircol™ Collagen Assay Kit. MCT model: Healthy, n=5, MCT, n=5; MCT+R428, n=6. SuHOX model: NOX, n=5;

SuHOX, $n=6$; SuHOX+R428, $n=7$. Data are presented as mean \pm SEM and statistical analysis was performed using One-way ANOVA.

-The authors show that treating PAECs in vitro with R428 increases ICAM-1 expression in PAECs. This might hamper barrier function and an explanation why the animals do worse. What is the impact of Axl modulation on barrier function?

- *This is an interesting point. To address this question, we implemented the FITC-dextran In Vitro Vascular Permeability Assay (ECM644, Millipore) in human PAECs. The experiments were performed in a 24-well receiver plate with 24 individual hanging cell culture inserts. As a positive control, TNF- α recombinant protein has been utilized. The R428 effects on PAEC permeability have been investigated in a time-dependent manner in comparison to DMSO-treated cells. Importantly, R428 treatment caused a significant increase in permeability of hPAECs in a time-dependent manner in comparison to DMSO-treated control cells. The new data are represented in Figure 2ⁱ. Please, see the new sentences included in the Results part. See Page 5, Lines 160-163. "Furthermore, R428 treatment of cultured hPAECs in a resulted in a significant time-dependent increase of endothelial monolayer permeability, as demonstrated through the use of a fluorescein isothiocyanate-dextran (FITC-dextran) transwell assay (Figure 2ⁱ)".*

- Axl has also been reported to modulate TGFbeta signaling. At 1 μ M, R428 is reported to prevent TGFbeta induced invation. As aberrant TGFbeta signaling is part of the PAH pathology, how is this pathway influenced and contribute to the phenotype observed? Is pSmad1/5 and pSmad2/3 influenced in the R428 treated animals and does this correlate with reduced Axl in those cells.

- *We followed the reviewer's suggestion and examined the phosphorylation status of Smad2/3 proteins in addition to the expression analyses of pSMAD1/5/8 in the total lung homogenates of R428- treated rats. Outstandingly, Axl inhibition strongly enhanced activation of both phospho-Smad2 and phospho-Smad3 proteins, indicating that Axl inhibition in vivo triggers a switch between TGF- β -Smad2/3 to BMPR2-Smad1/5/8 signaling. The new WB images and quantifications are now included in Figure 5. The following sentence has been incorporated into the Results part. Please see Pages 7-8, Lines 211-214: "The expression of phosphorylated SMAD2/3 proteins was markedly and significantly enhanced upon R428 treatment, indicating that Axl inhibition on PAH is linked with a switch between TGF- β -SMAD2/3 to BMPR2-SMAD1/5/8 signaling (Figure^{5d,e})". We have also implemented the correlation analyses between Axl and Bmpr2, Smad1/5/8, and pSMAD2/3 proteins. Our analyses show that Axl expression positively correlates with BMPR2 and pSmad1/5/8, but not with pSmad2/3. We have added the new analyses into Figure 5. The new text has been also included on Page 8, Lines 214-216: "In SuHx Lungs, Axl protein correlated positively with BMPR2 abundance (Figure^{5f}). Similarly, a highly significant correlation between Axl and pSMAD1/5/8, but not pSMAD2/3 was found in the Lungs of the SuHx rats (Figure^{5f})".*

- Figure 1 shows the effect of Axl inhibition on SMCs. In iPAH SMC the concentration needed of R428 is higher compared to control cells. Is this related to the expression level of Axl and the levels of BMPR2 in the cells.

We are thankful for this comment. We have determined the expression of BMPR2 in donor and IPAH hPASMCS and performed the correlation analyses between both proteins.

Indeed, *BMPR2* expression profile highly mimics the expression profile of the *Axl* receptor. Moreover, *Bmpr2* strongly correlates with *Axl* abundance in hPASMCs (Pearson $r=0.9576$, $P=0.0002$), i.e. has a high positive correlation. The new WB images and correlation analyses were incorporated into Supplemental Figure 8^{e,g}. The new sentences are included in the Results part of the manuscript. Please, see Page 10, Lines 264-267: "Interestingly, despite an elevation of *Axl* expression in lung homogenates of clinical PAH (Supplementary Figure 8^d), no changes in the expression profile of *Axl* protein were found in hPASMCs isolated from non-PAH individuals and IPAH patients (Supplementary Figure 8^e). Outstandingly, *Axl* demonstrates a highly significant positive correlation with *BMPR2* abundance in hPASMCs (Supplementary Figure 8^g)".

- In figure 2, the authors analyze the impact on EC behavior. These cells are extremely sensitive for R428. Is this related to *Axl* levels and *BMPR2* expression. Are IPAH patients ECs as sensitive?

-To address this question, we have performed several assays. First, we analyzed the expression profile of *Bmpr2* in human donor and IPAH PAECs. We noted that the *Bmpr2* protein expression was markedly and significantly reduced in IPAH-PAECs. Subsequently, we performed the correlation analyses between both proteins and found that *Axl* expression significantly correlates with *BMPR2* abundance in hPAECs (Pearson $r=0.7324$, $P=0.02$). WB image and graph demonstrating correlation is incorporated on Figure 8^{c-e}. The new sentence is included in the Results part of the manuscript. Please, see Page 10, Line 275-276: "A robust correlation between *Axl* and *BMPR2* protein expression in hPAECs was found (Figure 8^e)". Furthermore, we have repeated the cytotoxicity assays also in IPAH ECs. Importantly, R428 exhibited stronger impact on the toxicity of the diseased cells. The new data have been incorporated into Figure 2. Please, see the new sentences included in the Result part. Page 5, Lines: 145-147: "Remarkably, the IPAH-PAECs showed a stronger cytotoxic response in comparison to donor non-PAH PAECs (Figure 2^{b, c})."

- In figure 3, the authors show the impact of R428 on experimentally induced PH. Firstly, the figure legend is not correct, a-e is MCT and f-j is sugen. Secondly, the impact of *Axl* inhibition is more clear in MCT compared to SuHOX. Can the authors explain this? Does this reflect hypoxia induced vs inflammation/liver induced disease progression? What would be most predictable for the patient?

- We are thankful for noticing this. The Figure Legend was corrected. We believe that in the MCT model of PAH, which is characterized by inflammation, *Axl* inhibition further enhances the pro-inflammatory signaling. This is consistent with previous reports in which *Axl* via *Gas6* was demonstrated to control the inflammation, while R428 antagonized the anti-inflammatory effect of *Gas6*. Furthermore, *Axl* inhibition upon MCT administration resulted in obliteration of the small pulmonary arteries, the not typical for the MCT model of PAH. Thus, we assume that *Axl* blockage directly triggers ECs, causing their damage, partially resembling the severe SuHx model of PAH. In the setting of SuHx model, *Axl* inhibition probably extends the initial period of EC predisposition to apoptosis, which markedly affects a secondary hyper-proliferative EC phenotype culminating in PAH.

- In figure 4, the impact of R428 in SuHOX lungs is analyzed in more detail. Which cell type is affected?

-We assume that various cell types may be affected in SuHOX Lungs. Since there is the unique composition of various markers of inflammation and apoptosis, we suspect that the most affected by R428 cell type in the Lungs are ECs (since no apoptosis have been noted in SMCs of pulmonary vessels and a strong drop in the number of PMVECs and PAECs were noted). Another cell type, which we believe might be strongly affected in this disease setting by Axl inhibition is pro-inflammatory CD68 positive macrophage.

- The authors nicely show that R428 modulates BMPR2 levels in lung tissue of suHOX animals. Recent studies have shown that activin and activin receptors are involved in PAH. How does R428 influence this pathway?

We have performed WB analyses of ActR-2A and ActR-2B in Lung tissues of SuHOX rats. Our analyses reveal that Axl inhibition does not modulate the expression profile of both receptors, indicating that the effect of Axl is more specific on Bmpr2 signaling. New data are incorporated into Supplemental Figure 7, and the following sentences were added into the text of the Results part. Please, see Page 8, Lines 216-219: "Outstandingly, WB analyses of Lung protein extracts indicated that Axl inhibition does not impact the expression of Activin A receptor type 2A (ActR-2A) and Activin A receptor type 2B (ActR-2B) proteins (Supplemental Fig^{7a, b}).

- In the last part of the paper, the authors aim to unravel the mechanism of action, and suggest that Axl interacts with BMPR2. In figure 7/8 the authors show that Axl and R428 interfere with BMP/SMAD1/5 signaling by modulating BMPR2 levels. This is an interesting mechanism, but this reviewer has some concerns. The effect seen treating the cells with R428 could very well be indirect. R428 induces a proinflammatory phenotype, demonstrated by the high iCAM and high IL6. Previous work, also by some of the co-authors, have demonstrated that this eventually results in BMPR2 downregulation. Therefore more solid validation is needed to demonstrate that this is a direct effect.

In our experiments, we demonstrate that Axl deletion by a siRNA-mediated approach resulted in an inhibition of Gas6 and Bmp9 induced Smad1/5/8 phosphorylation and ID1 and ID2 decline, indicating that Axl deficiency plays a detrimental role in hPAECs. Various assays in which R428 was employed (Caspase activation, E-selectin expression) support a notion of a protective impact of Axl on EC dysfunction. Immunoprecipitation assays in hPAECs revealed growth-factor-induced interaction of both receptors, indicating that both receptor signaling pathways interfere. R428 induces a pro-inflammatory phenotype in experimental PAH in vivo. Furthermore, the knock-down of Axl in hPAECs resulted in ICAM1 augmentation, but VCAM1 decrease, suggesting that small molecular inhibitor of Axl signaling pathway and Axl deletion might contribute to non-overlapping molecular signaling pathways.

1. The concentration of BMP9 and Gas6 used to stimulate the cells is very high. If Gas6 is contaminated with some kind of BMP, you can have the results shown. Therefore, a neutralizing antibody should be used in combination with Gas6.

-We are very thankful for this comment. The concentration of Gas6 (200ng/ml) was chosen based on previous publications. For example, Gas6 was employed in cancer cells at the doses of 200-400ng/ml (PMID: 15605394, PMID: 19888345, PMID: 11154277). Gas6 was utilized at 200ng/ml in human aortic SMCs (PMID: 9507025). In human aortic ECs

Axl activation was achieved by Gas6 at 100-250ng/ml of (PMID: 27006397). At the doses of 200-400ng/ml Gas6 affected tube formation and wound-healing cell migration of human retinal ECs (PMID: 24409287). To assure the specificity of the Gas6 stimulation, we as recommended by the reviewer, pre-treated hPAECs with the anti-Gas6 neutralizing antibody (human Gas6 goat antibody (AB885) (50µg/ml)) or with the normal goat IgG control (AB-108-C) (50µg/ml), both provided by RandD, as previously reported (PMID: 31311882). We have repeated experiments, in which hPAECs were shortly stimulated with Gas6 recombinant protein for Smad1/5/8 activation. Importantly, anti-Gas6 neutralizing antibody completely abrogated the action of Gas6 on Smad1/5/8 activation in short-term stimulations. Please see the new data included into Supplemental Fig^{8b}). The following sentence has been incorporated into the text of the manuscript. Please, see Page 9, Lines 249-251: "Importantly, Gas6 neutralizing antibody completely obstructed Gas6-mediated SMAD1/5/8 activation (Supplementary Figure ^{8b}) and BMPR2 increase (Supplementary Figure ^{8c})". Moreover, we have implemented long-term (48h) experiment, in which hPAECs were stimulated for 48h with Gas6 recombinant protein. These experiments were also controlled by the pre-treatment of Gas6 neutralizing antibody.

2. Neither BMP9 nor Gas6 were able to induce BMPR2 expression in fig 7K

-We would like to mention that all stimulations in fig 7k have been performed in a short-term manner. Specifically, hPAECs were starved in low-serum medium and after o/n pre-treated with R428 inhibitor for 1.5hours. After drug pre-treatments, cells were stimulated with the indicated doses of BMP9 and Gas6 diluted in the medium containing 0.2%FCS for 2hours (please, see the Figure Legends of Figure 7^{k-m} for details). To confirm that Gas6, similarly to BMP9 stimulation, enhances BMPR2 expression, the long-term (48h) stimulations in the presence of neutralizing antibody were also performed. Importantly, Gas6 stimulations enhanced BMPR2 accumulation, which was completely abolished in the presence of Gas6Ab. Please, see the new data included in Supplementary Figure ^{8c}.

and Id1 and pSmad1/5 should be done in the presence of siRNA for BMPR2 and ACVR2A/B.

-We have performed new experiments, in which we analyzed the expression of phospho-Smads (protein) and ID1/ID2 (mRNA) upon siRNA-mediated deletion of both Axl and Bmpr2. The deletion of both receptors in hPAECs resulted in attenuation of growth-factor induced Smad pathway activation reciprocally, as determined by the decrease of Smad phosphorylation and the decrease of ID1 and ID2 expression. The new data has been incorporated into Figure 7. Please, see Figure ^{7n-p} for details. The following sentence has been incorporated in the text of the manuscript. Please, see Page 9, Lines 256-257: "The knock-down of Axl and BMPR2 each diminished BMP9- and Gas6-induced SMAD1/5/8 activation both directly and reciprocally (Figure ^{7n,o})". Furthermore, a new qPCR analysis of ID1/ID2 expression was incorporated in Figure 7. Please, see the new Figure ^{7p} and the following sentences on Page 9, Lines 256-260. "A significant reduction of ID1 and ID2 mRNA transcripts was also noted in Axl small interfering RNA-treated hPAECs, indicating that Axl is required for BMP signaling transduction to Smad1/5/8 phosphorylation and ID1/ID2 increase in ECs (Figure ^{7o-q}). Similarly, BMPR2 knock-down lessened Gas6 prompted augmentation of ID1/ID2 expression, suggesting an interaction of both pathways."

Unfortunately, we were not successful with the experiments targeting both ACVRA/b in hPAECs, as the antibodies detecting both ACVRA and ACVRb in human ECs did not provide confident results.

3. Figure 7G lacks a BMP control.

-We have newly performed the experiments with the addition of BMP9 control stimulations. The new data have been incorporated into Figure 7^g and the corresponding sentences have been included in the Figure Legends Part of the manuscript. Please, see Page 40, Line 845.

4. In figure 8, to show interaction, the IP performed should be repeated with BMP9 and Gas6 to draw this conclusion.

-We have repeated the IP experiments in which hPAECs were stimulated for 48h by growth factors. New data has been incorporated into Figure 8 (please, see Figure 8^j). The following sentence has been added to the text of the manuscript. Please, see Pages 10-11, Lines 283-285: "Both immunoprecipitations and proximity ligation assays exhibited enhanced receptor-receptor interaction upon simultaneous BMP9/Gas6 stimulations of hPAECs (Figure 8^{j-l})".

5. The PLA signals in fig 8i is not clear. Not all signal seems to be at the cell membrane.

-We have adjusted the brightness of all images in Figure 8a. We believe, that the appearance of the dots in the cytoplasm of the hPAECs might be due to the internalization of the receptors, which is much more remarkable upon growth factor stimulations.

Minor comment

- As not all samples are shown by WB, it would be informative if dots were shown in the bar graph as was done in figure 3.

- We are thankful for this comment. Western Blots analyses representing individual animals, as well as cell culture experiments are now represented as dots. Please, see changed main Figures (Fig-s 1,2,4,5,6,7,8) and Supplementary Figures (1, 4, 5, 6, and 7) for details.

- Not all legends show the number of samples analyzed.

- Necessary changes have been performed.

- Molecular weights need to be indication for the western blots.

- Molecular weights were incorporated adjacent to all WB images.

- Suppl figure 5 A: pStat3/ stat3 and pan-actin only have 13 lanes while Nitro and NFkB have 14 lanes.

-We have adjusted Figure 5.

- Suppl Figure 6C, pan actin has 14 lanes while pdl1 only 13.

-We have adjusted Figure 6.

- Suppl figure 6H: from the images shown, there is more proliferation in SuHOX with R428.

-The image (SuHOX+R428) was replaced with another one.

Reviewers' comments:

Reviewer #2 (Remarks to the Author):

I would like to thank the authors by answering almost all of my questions, and I can agree with the inability to perform additional animal experiments. This paper will have an impact on PAH research.

There is one minor issue that needs attention.

1. The authors analyzed TGFbeta signaling upon R428 administration using WB analysis. The authors indicated the size of the proteins. While indeed pSmad3 is approx. 52 kDa, pSmad2 has a height of 60 kDa. Are the authors sure to have the correct band? Was there a positive control run along to indicate the size?

2. While the increase in pSmad3 is clear, the increase in pSmad2 is mainly due to reduced levels of Smad2, suggesting that there is sufficient signaling to activate downstream gene transcription in all conditions. What is the increase of pSmads when correlated to e.g. pan Actin?

3. Why is vinculin in figure 5D?

4. In the model presented the authors depict Id1 and Id2. As Id3 is often used, being the same type of co-factor and present in ECs, what is the rationale for Id2 and not Id3?

Reviewers' comments:

Reviewer #2 (Remarks to the Author):

I would like to thank the authors by answering almost all of my questions, and I can agree with the inability to perform additional animal experiments. This paper will have an impact on PAH research.

There is one minor issue that needs attention.

1. The authors analyzed TGFbeta signaling upon R428 administration using WB analysis. The authors indicated the size of the proteins. While indeed pSmad3 is approx. 52 kDa, pSmad2 has a height of 60 kDa. Are the authors sure to have the correct band? Was there a positive control run along to indicate the size?

1. The reviewer is correct. The pSmad2 should be run at 60kDa. However, the only band which was detected in our WBs with pSmad2 antibody was the band slightly above 50kDa (see the data source file for details). To assure that the band is correct, we run the freshly prepared protein lysates and re-probed the membranes with a freshly prepared pSmad2 antibody. Simultaneously with these samples, we run on the separate gel the protein extracts isolated from hPAECs, after BMP9/Gas6 stimulations (these experiments have been previously performed to determine the effect of R428 on Smad2/Smad3 activation). Interestingly, in both of these membranes the only band detected for pSmad2, was the one located at the same height (above the 50kDa of the marker). BMP9, as reported before, induced Smad2 activation in hPAECs. As, cells stimulated with BMP9, provided similar outcome, we assume that the band which we detected by our WB analyses was the right one. Please, see the image below for details.

To be more precise we exchanged the mark on the WB image for both pSmad2 and Smad2 on Figure 5. Now we define the position of the marker on the WB image for both pSmad2/Smad2. Please, see the new Figure 5 for details.

2. While the increase in pSmad3 is clear, the increase in pSmad2 is mainly due to reduced levels of Smad2, suggesting that there is sufficient signaling to activated downstream gene transcription in all conditions. What is the increase of pSmads when correlated to e.g pan Actin?

As suggested by the reviewer we first quantified the expression of pSmad2 and pSmad3 to the Pan-actin. We found that there is indeed insignificant upregulation of pSmad2 levels in comparison to Pan-actin, while pSmad3 normalized to Pan-actin was found to be significantly induced. To substantiate the results, we normalized the levels of Smad2 to Pan-actin first and then performed new quantification of pSmad2 to the normalized Smad2 protein. The quantification analyses indicated that indeed pSmad2

though augmented by R428 treatment in comparison to NOX controls, was not significantly altered in comparison to SuHOX+Placebo treated controls. Thus, we exchanged both the Figure 5 graph (see the new graph included), as well as we provide better quality images of new pSmad2, Smad2 (see the image above), and the Pan-actin, related to this membrane. Similarly, we quantified pSmad3 to tSmad3, after normalization to Pan-actin. Here, as expected, no difference in comparison to the previously shown data was noted. Both new graphs for quantification of pSmad2 and pSmad3 are incorporated into Figure 5. Accordingly, we modified the sentence on the Results Part. Please, see Pages 7-8, Lines 211-213, in which we removed the word "**significantly**:" The expression of phosphorylated SMAD2/3 proteins was markedly enhanced upon R428 treatment, indicating that Axl inhibition on PAH is linked with a switch between TGF- β -SMAD2/3 to BMPR2-SMAD1/5/8 signaling (Figure 5d, e). Consequently, we also substituted the graphs for correlation analyses of Axl/pSmad2 and Axl/pSmad3 (See Figure 5f for details).

3. Why is vinculing in figure 5D?

Both BMPR2 and Axl possess high molecular weight (the size for both proteins is above 100kDa), while ID1 protein size is comparatively low (15kDa), thus for better separation and cathing of all the bands, several WBs were running. The ID1 detection was performed on a different gel, than Bmpr2 and Axl. For ID1 protein normalization vinculin was used, as a loading control. Thus we included this WB image in Figure 5 too. We have added the correct labeling for panel 5 on the y-axis (ID1) to the bar graph.

4. In de model presented the authors depict Id1 and Id2. As Id3 is often used, being the same type of co-factor and present in ECs, what is the rationale for Id2 and not Id3?

According to our knowledge all ID1, ID2, and ID3 represent a complex of transcriptional factors downstream of BMPR2 signaling. The transcriptional regulators, ID1 and ID2, were established as are downstream targets of canonical BMP signaling that promote cell proliferation (PMID: 11840324). It was previously reported, that Id1 and Id3 serve as critical downstream effectors of BMP signaling in PASCs [PMID: 23771984], as the loss of BMPR2 function reduces the induction of ID1 and ID3 genes in PASCs, affecting the proliferation of PASCs via cell cycle inhibition. We were specifically interested in the anti-apoptotic role of the BMP9/BMPR2 and the anti-apoptotic impact of this axis in hPAECs. Thus, our experimental settings were based on the report suggested that BMP9 signals through BMPR2 to induce the transcription of Id1, Id2, E-Selectin, IL-6, and IL-8 in ECs [PMID: 19366699]. Furthermore, one more recent report also indicated that the endothelial-selective BMPR-II ligand, BMP9 resulted in an early induction of both *ID1* and *ID2* in PAECs. Thus, for our analyses, ID1 and ID2 were chosen.